# Enhancing alphafold-multimer-based protein complex structure prediction with MULTICOM in CASP15

Jian Liu[1], Zhiye Guo [1], Tianqi Wu[1], Raj S. Roy[1], Farhan Quadir[1], Chen Chen[1] & Jianlin Cheng [1✉]

To enhance the AlphaFold-Multimer-based protein complex structure prediction, we developed a quaternary structure prediction system (MULTICOM) to improve the input fed to AlphaFold-Multimer and evaluate and refine its outputs. MULTICOM samples diverse multiple sequence alignments (MSAs) and templates for AlphaFold-Multimer to generate structural predictions by using both traditional sequence alignments and Foldseek-based structure alignments, ranks structural predictions through multiple complementary metrics, and refines the structural predictions via a Foldseek structure alignment-based refinement method. The MULTICOM system with different implementations was blindly tested in the assembly structure prediction in the 15th Critical Assessment of Techniques for Protein Structure Prediction (CASP15) in 2022 as both server and human predictors. MULTICOM_qa ranked 3rd among 26 CASP15 server predictors and MULTICOM_human ranked 7th among 87 CASP15 server and human predictors. The average TM-score of the first predictions submitted by MULTICOM_qa for CASP15 assembly targets is ~0.76, 5.3% higher than ~0.72 of the standard AlphaFold-Multimer. The average TM-score of the best of top 5 predictions submitted by MULTICOM_qa is ~0.80, about 8% higher than ~0.74 of the standard AlphaFold-Multimer. Moreover, the Foldseek Structure Alignment-based Multimer structure Generation (FSAMG) method outperforms the widely used sequence alignment-based multimer structure generation.

[1]Department of Electrical Engineering and Computer Science, NextGen Precision Health Institute, University of Missouri, Columbia, MO 65211, USA.
✉email: chengji@missouri.edu

Single-chain proteins (monomers) often interact with each other to form multimers (i.e., assemblies or complexes) to perform functions such as gene regulation and signal transduction. The quaternary structures of multimers largely determine their function. Therefore, it is important to predict the quaternary structure of protein complexes from their sequences for studying protein-protein interaction and function. However, predicting the quaternary structure of protein complexes is more difficult than predicting the tertiary structure of single-chain monomers because the former involves more than one protein chain and needs to consider both intra-chain residue-residue interaction and inter-chain residue-residue interaction.

Traditionally, the prediction of protein complex structures employs template-based modeling or ab initio methods such as protein-protein docking. In template-based modeling, complex templates with known structures are initially identified for a target protein complex and subsequently utilized to construct its structural prediction. While this approach proves effective if similar homologous templates can be found, it does not work for most targets because good templates are usually not available or cannot be identified.

In contrast, ab initio methods aim to predict complex structures without the reliance on templates, utilizing various techniques such as grid-based fast-Fourier transform docking[1–3], particle swarm optimization for conformational sampling[4], and local shape complementarity alongside symmetry constraints[5]. Furthermore, integrative methods[6,7] combine template-based modeling and ab initio docking to improve the prediction of complex structures. Nevertheless, the accuracy of these methods for predicting complex structures is generally low[8,9], due to the absence of templates, complexities associated with conformational sampling, inaccuracy of scoring functions, and challenges in accommodating protein flexibility.

The recent application of deep learning to inter-protein contact prediction and quaternary structure prediction has started to transform the field[10–15]. Particularly, the adaption of the high-accuracy tertiary structure prediction method—AlphaFold2[16]—for quaternary structure prediction as AlphaFold-Multimer[13] has drastically improved the accuracy of quaternary structure prediction for protein assemblies. AlphaFold-Multimer is an end-to-end protein complex structure prediction method whose accuracy depends mostly on the quality of multiple sequence alignment (MSA) input, even though good structural templates may have some positive effect.

Despite the breakthrough made by AlphaFold-Multimer, its accuracy for quaternary structure prediction is still much lower than AlphaFold2's accuracy for tertiary structure prediction. Therefore, there is still a large room to further improve the accuracy of AlphaFold-Multimer-based complex structure prediction.

In this work, we developed several algorithms to improve AlphaFold-Multimer-based complex prediction from different aspects and integrated them to build a MULTICOM complex structure prediction system. It uses both traditional sequence alignments and Foldseek[17]-based structure alignments to generate MSAs for monomers and concatenates them as MSAs for multimers according to different criteria such as the same species and known/hypothetical protein-protein interaction. The structural templates identified by the sequence or structure alignments for monomers from different template databases are also combined as templates for the multimers. The diverse set of MSAs and templates are used as input for AlphaFold-Multimer to generate quaternary structural predictions, which are then ranked by multiple complementary quality assessment (QA) methods including AlphaFold-Multimer's confidence score, the average pairwise structural similarity (PSS) between a prediction and

other predictions of the same target, and the average of the two. The top-ranked predictions are further refined by using the Foldseek structure alignment-based multimer structure refinement to generate better predictions.

We implemented the MULTICOM system as two server predictors and two human predictors that blindly participated in the assembly structure prediction in CASP15 from May to August 2022. Both the MULTICOM server and human predictors ranked among the top server or human/server predictors in CASP15. The predictors also performed significantly better than a standard AlphaFold-Multimer predictor participating in CASP15, demonstrating that the MULTICOM approach has significantly improved the accuracy of the AlphaFold-Multimer-based protein assembly structure prediction. We released the source code of the MULTICOM system at GitHub so that the community can run it on top of AlphaFold-Multimer to obtain more accurate protein complex structure predictions.

## Results and discussion

**The comparison between MULTICOM servers and other CASP15 server assembly predictors.** According to the CASP15 official assessment (see the official ranking https://predictioncenter.org/casp15/zscores_multimer.cgi), MULTICOM_qa and MULTICOM_deep servers ranked 3rd and 5th among all CASP15 assembly server predictors. The MULTICOM human predictors (MULTICOM_human and MULTICOM) ranked 7th and 10th among all CASP15 assembly predictors. The official CASP15 ranking metric (https://predictioncenter.org/casp15/doc/presentations/Day2/Assessment_Assembly-CASP_EKaraca.pdf) to score a prediction in a pool of predictions for a target is $\frac{Zscore_{ICS} + Zscore_{IPS} + Zscore_{TM-score} + Zscore_{lDDTo ligo}}{4}$, which is the average of Z-scores of ICS (Interface Contact Score)[18], IPS (Interface Patch Score)[18], TM-score calculated by US-align[19] and lDDToligo (Oligomeric lDDT)[20]. Such a score was calculated for the no. 1 prediction for each target submitted by each predictor. The sum of all positive Z-scores for all the CASP15 targets is the total score of a predictor, which is used to rank all the predictors as shown in Table 1. In addition to the top 1 submitted predictions, CASP15 also calculated the total score for the best of five predictions for the targets submitted by a predictor to rank the predictors alternatively. According to the Z scores in terms of the best of five predictions, MULTICOM_deep, MULTICOM_qa ranked 2nd and 3rd among 26 server predictors as shown in Table S1. MULTICOM_human and MULTICOM ranked 10th and 14th among 84 predictors.

The average TM-scores of top 1 predictions (or best of five predictions) submitted for 41 multimer targets by the top 15 CASP15 server predictors including the standard AlphaFold-Multimer (i.e., NBIS-AF2-multimer run by the Elofsson Group) according to the CASP15 official Z-score ranking are reported in Table 1 (or Table S1). The TM-score of a prediction is calculated by using US-align[19] with parameters (-TMscore 6 -ter 1) to compare it with the native structure. 41 multimeric targets include 20 hetero-multimers and 21 homo-multimers. A target is classified as a template-based modeling (TBM) target if a rather complete template could be found for it and its subunits, while a target is classified as free-modeling (FM) or FM/TBM target if no template or only a partial template could be found for it or its subunits. Out of 41 multimeric targets, 14 of them are classified as TBM targets, 27 of them are classified as FM or FM/TBM targets.

The average TM-score of top 1 predictions submitted by our best MULTICOM server predictor (MULTICOM_qa) for the 41 multimers is 0.7565, only slightly lower than the highest score 0.7665 of the no. 2 Z score ranked server predictor Manifold-E. It is worth noting that CASP15 Z score-based ranking is not the

**Table 1 The top 15 out of 26 server predictors including NBIS-AF2-multimer ranked by the CASP15 official Z-score and the average TM-score of top 1 predictions submitted by them for the 41 multimers, 14 TBM multimers, 27 TBM/FM and FM multimers.**

| Server predictors | Sum of $Z$ scores (> 0.0) | Target count | Avg TM-score on 41 multimers | Avg TM-score on 14 TBM multimers | Avg TM-score on 27 FM and FM/TBM multimers |
|---|---|---|---|---|---|
| Yang-Multimer[42] | **24.6946** | 38 | 0.7138 | 0.8235 | 0.6569 |
| Manifold-E[43] | <u>18.8589</u> | 41 | **0.7665** | 0.8211 | **0.7382** |
| MULTICOM_qa[44] | 18.3529 | 41 | <u>0.7565</u> | 0.8111 | <u>0.7281</u> |
| DFolding-server[45] | 17.0135 | 33 | 0.5978 | 0.6634 | 0.5637 |
| MULTICOM_deep[44] | 16.2869 | 41 | 0.7416 | **0.8459** | 0.6875 |
| UltraFold_Server[46] | 15.7081 | 41 | 0.6961 | 0.7884 | 0.6483 |
| MultiFOLD[47] | 15.2358 | 41 | 0.6643 | 0.7366 | 0.6268 |
| MUFold[48] | 14.0905 | 41 | 0.7195 | <u>0.8401</u> | 0.6569 |
| Kiharalab_Server[49] | 13.5184 | 40 | 0.6703 | 0.7597 | 0.624 |
| ColabFold[50] | 12.7694 | 39 | 0.6339 | 0.7148 | 0.592 |
| NBIS-AF2-multimer[51] | 12.271 | 41 | 0.7186 | 0.8163 | 0.668 |
| RaptorX-Multimer[52] | 11.9178 | 40 | 0.6744 | 0.78 | 0.6196 |
| Yang-Server[42] | 10.495 | 19 | 0.3578 | 0.6174 | 0.2232 |
| DFolding-refine[45] | 9.3295 | 36 | 0.6584 | 0.8144 | 0.5775 |
| GuijunLab-Assembly[53] | 8.6 | 41 | 0.6701 | 0.769 | 0.6188 |

When calculating the average TM-score here, if a predictor did not submit a prediction for a target, the TM-score for the target is set to 0.
The bold font highlights the best result. The underline denotes the second-best result.

same as the average TM score-based ranking because the former favors the predictors that perform well on some targets when most other targets fail, which is different from the latter weights all the targets equally. This is the reason Yang-Multimer ranked no. 1 in terms of Z-score even though it missed three targets. The average TM-score of top 1 predictions of MULTICOM_qa is 5.27% higher than 0.7186 of NBIS-AF2-multimer, showing that a pronounced improvement has been made over the standard AlphaFold-Multimer. Like all the other server predictors, MULTICOM_qa and MULTICOM_deep performed better on the TBM targets than on the FM and FM/TBM targets. MULTICOM_deep has the highest average TM-score of 0.8459 on the TBM targets. MULTICOM_qa has the second highest average TM-score of 0.7281 on the 27 FM and FM/TBM targets, only lower than 0.7382 of Manifold-E.

MULTICOM_qa performed obviously worse than the top-ranked server predictors—either Manifold-E or Yang-Multimer—(i.e., TM-score difference >0.08) on three homo-trimers (T1174o, T1179o and T1181o), two large hetero-multimers (H1114 and H1137), three hetero-dimers including two nanobodies (H1141 and H1144) and an antibody-antigen H1129, and a large homo-multimers (T1176o) (see Fig. 1). For T1174o, T1179o and T1181o, MULTICOM_qa managed to generate some good predictions in the prediction pool, but the ranking method failed to select them as top 1 prediction. For H1114 (stoichiometry: A4B8C8), because there was no sufficient GPU memory for AlphaFold-Multimer to generate full-length predictions for the entire complex of 7,988 residues, MULTI-COM_qa tried to predict the structures for different components of the complex (e.g., ABC, AB2C2, A3B3C3, AB4C2, B8) and then combined them to build the structure for the entire complex. Unfortunately, it did not try to build the structure of the A4 component, which is the key to linking all the components together. Therefore, MULTICOM_qa submitted a structure predicted for AB2C2 as the top 1 prediction as shown in Fig. 1.

For H1137 (stoichiometry: A1B1C1D1E1F1G2H1I1), MULTI-COM_qa predicted two conformations for the six-chain trans-membrane helical channel consisting of Chains A, B, C, D, E and F (one relatively straight one and one bended one), but it selected the less accurate bended one according to the pairwise similarity between predictions, leading to the mediocre quality of predicted structures for the target (see Fig. 1).

For two nanobody-antigen complexes H1141 and H1144 (stoichiometry: A1B1), there are different reasons for the failure. For H1141, the maximum TM-score of the predictions generated by MULTICOM is 0.6838, much lower than 0.96 of the top 1 prediction submitted by Manifold-E. For H1144, although some good structural predictions (TM-score = ~0.89) had been generated, the ranking method selected a common conformation of low quality rather than the high-quality predictions that were rare in the prediction pool. For nanobody targets like H1141 and H1144, it would be useful to generate a large number of predictions to obtain more high-accuracy predictions that may have obviously higher confidence scores than other low-quality predictions. Moreover, as nanobody targets do not necessarily have inter-chain co-evolution information recorded in their MSAs, it may be useful not to pair their MSAs when using AlphaFold-Multimer to generate predictions for them as shown by some predictors in CASP15.

For H1129, our monomer alignments pairing method did not find any pairs for the two subunits. Therefore, only several default MSAs and template combinations (e.g., default_multimer, default_pdb, default_pdb70, default_comp, default_struct, default_af) were used as inputs for AlphaFold-Multimer to generate 30 predictions for it. The maximum TM-score of the generated predictions is 0.8149, much lower than 0.964 of the top 1 prediction submitted by Yang-Multimer. T1176o is a homo-multimer with 8 subunits that interact via multiple interfaces, for which AlphaFold-Multimer could generate full-length complex structures directly. However, the interaction between the 8 sub-units could not be well predicted by AlphaFold-Multimer. Predicting the structure for 2 or 3 units (A2 or A3) of the multimer produced several different conformations and inter-faces, which could not be easily combined to generate good full-length predictions for the complex. In fact, no prediction submitted by all the CASP15 predictors have a TM-score >0.5, indicating this is a very hard target.

**Overall performance of MULTICOM_qa compared with the standard AlphaFold-Multimer.** Fig. 2a shows the distribution of TM-scores of the best of five predictions submitted by MULTI-COM_qa on the 41 multimers (14 TBM multimers and 27 TBM/FM and FM multimers). For 31 out of 41 (75.61%) targets, it

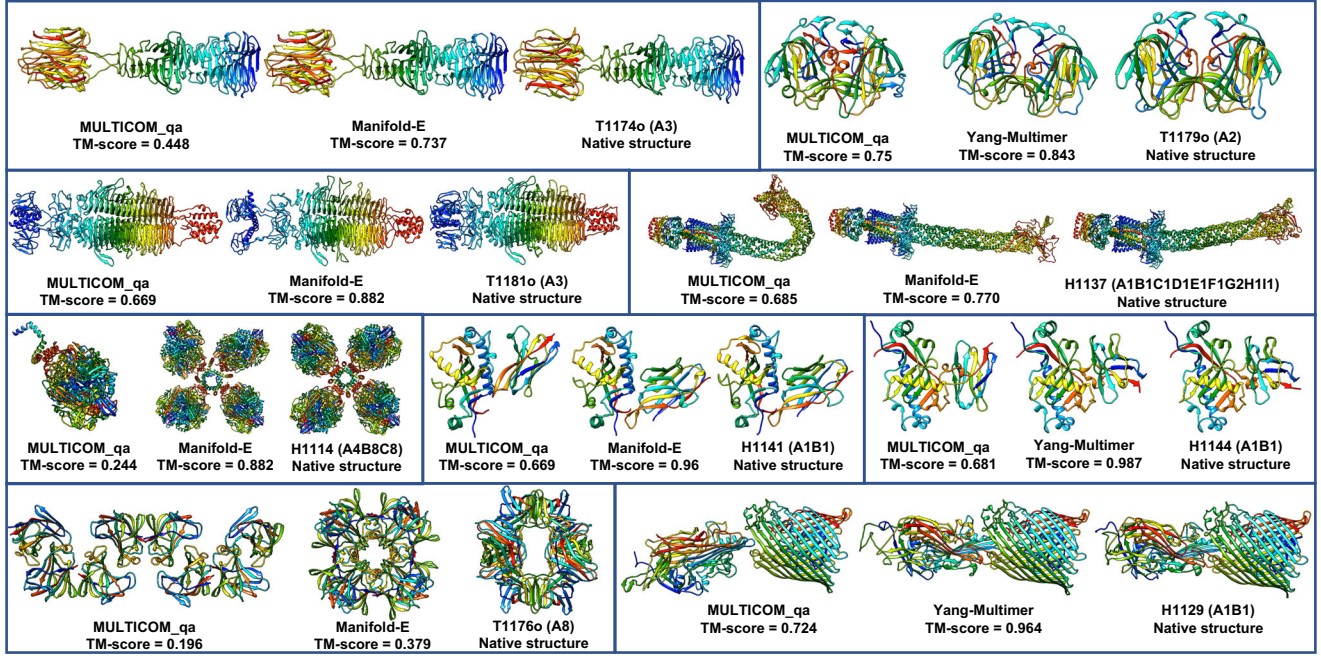

**Fig. 1 The targets on which MULTICOM_qa underperformed.** The top 1 submitted predictions for 9 targets on which MULTICOM_qa performed worse than Manifold-E or Yang-Multimer.

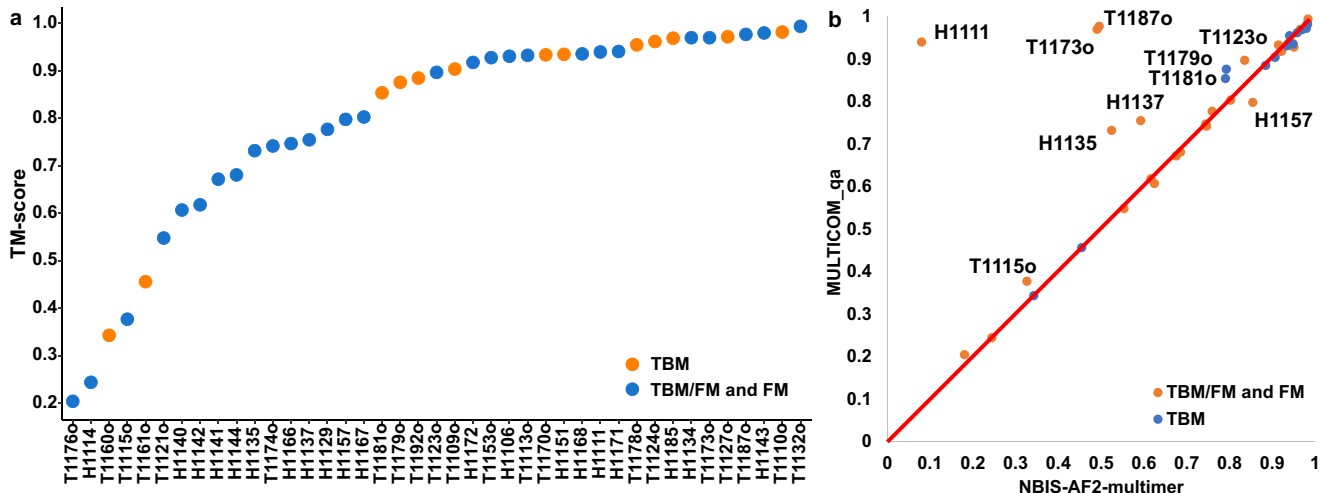

**Fig. 2 Comparative analysis of MULTICOM_qa and NBIS-AF2-multimer on 41 multimer targets. a** The plot of the best TM-scores of the top 5 predictions submitted by MULTICOM_qa on the 27 TBM/FM and FM targets, and 14 TBM targets in the increasing order. The per-target average TM-score of the best prediction is 0.7963 on the 41 multimer targets. **b** The plot of the TM-score of the best of the top 5 predictions submitted by MULTICOM_qa for each target against that of NBIS-AF2-multimer on 41 multimer targets.

generated at least one prediction with TM-score >= 0.7. For 24 out of 41 (58.54%) targets, it generated at least one high-quality prediction with TM-score > 0.85.

However, MULTICOM_qa performed very poorly on H1114, T1176o, T1115o, T1160o and T1161o (TM-score of the best prediction <0.5) as shown in Fig. 2a and Fig. S1. H1114 (stoichiometry: A4B8C8), T1176o (stoichiometry: A8) and T1115o (stoichiometry: A16) are very large multimers. The reason why MULTICOM_qa failed on H1114 and T1176o is explained in Section 2.1. T1115o is a large homo-multimer with 16 subunits (4608 residues), for which AlphaFold-Multimer could not generate full-length complex structures due to the lack of GPU memory. Therefore, MULTICOM_qa tried to predict the multimer structures for A4 and A8, which were combined into an

arc-like structure for the complex whose bending angles were different from the native structure.

The two homodimers (T1160o and T1161o) have two very short chains (48 residues only). They have very similar sequences (only five-residue differences in the sequence of the chain) but fold into two different conformations due to different crystallization conditions, which may make it harder for AlphaFold-Multimer to predict their structures. In fact, few CASP15 predictors made good predictions for these two targets, even though AlphaFold-Multimer assigned very high confidence scores (e.g., >0.8) to the incorrect predictions, indicating they may be outliers.

Figure 2b compares the TM-score of the best of the top 5 predictions that MULTICOM_qa submitted for each of 41

multimers against that submitted by the standard AlphaFold-Multimer—NBIS-AF2-multimer. On almost all the targets, MULTICOM_qa was able to generate predictions with quality better than or similar to NBIS-AF2-multimer. Particularly, MULTICOM_qa performed substantially better (e.g., TM-score difference >0.05) than NBIS-AF2-multimer on nine targets (H1111, T1187o, T1173o, H1135, H1137, T1179o, T1181o, T1123o and T1115o), while NBIS-AF2-multimer only has an obviously higher score (e.g., difference >0.05) than MULTICOM_qa for only one target (H1157). The average best TM-score of the top 5 predictions of MULTICOM_qa on the 41 multimer targets is 0.7963, which is about 8.0% higher than 0.7375 of NBIS-AF2-multimer. The p-value of the difference is 0.026 according to one-sided Wilcoxon signed-rank test. Fig. 3 illustrates the nine examples on which MULTICOM_qa substantially outperformed NBIS-AF2-multimer.

**Sampling predictions with diverse MSAs and templates improves assembly structure prediction.** We compare the best prediction generated from the MSA-template combinations in Table S2 and Table S3 with that of NBIS-AF2-multimer on each of 31 out of 41 CASP15 assembly targets. 10 targets are not included into this analysis for several reasons: unavailability of native structures for T1115o, T1192o, and H1185, multiple structural conformations for H1171 and H1172, and no or few full-length structures (i.e, <5 predictions) generated for H1111, H1114, H1135, H1137 and T1176o directly by the customized AlphaFold-Multimer in the MULTICOM server system during CASP15 because there was no sufficient GPU memory. The top 5 predictions generated by the MULTICOM server system for the

31 targets are selected according to their AlphaFold-Multimer confidence scores.

Figure 4 compares the TM-score of the best of the five predictions predicted from the diverse MSAs and templates generated by the sequence alignment component in the MULTI-COM server system against that of NBIS-AF2-multimer on the 31 multimer targets. The average TM-score of the best predictions generated by the MULTICOM server system is 0.813, higher than 0.789 for NBIS-AF2-multimer. The results demonstrate using diverse MSAs and templates generated by different sequence alignment approaches as input for AlphaFold-Multimer to generate more predictions can improve the quality of the best possible predictions over the standard AlphaFold-Multimer.

Increasing the value of AlphaFold-Multimer parameter num_ensemble_eval from 1 to 3 and num_recycle from 3 to 5 and updating the sequence and template databases to the time slightly prior to the start date of CASP15 can also slightly improve the quality of the predictions generated. For instance, the average per-target best TM-score of using AlphaFold-Multimer with the updated databases and adjusted parameters is 0.8013, slightly higher than 0.789 of NBIS-AF2-multimer. It is worth noting that the sequence databases of NBIS-AF2-multimer were also updated to April 2022 and its template database was updated to May 2022.

To further analyze the performance of all 13 kinds of MSAs in Table S4 on protein complexes, after CASP15 was concluded, we created a benchmark dataset[21] from the proteins deposited in the PDB after AlphaFold-Multimer was released as follows. We retrieved complex structures from the PDB released between 04/01/2022 and 12/09/2022. The structures were subjected to a series of filtering, considering the criteria[13] such as sequence

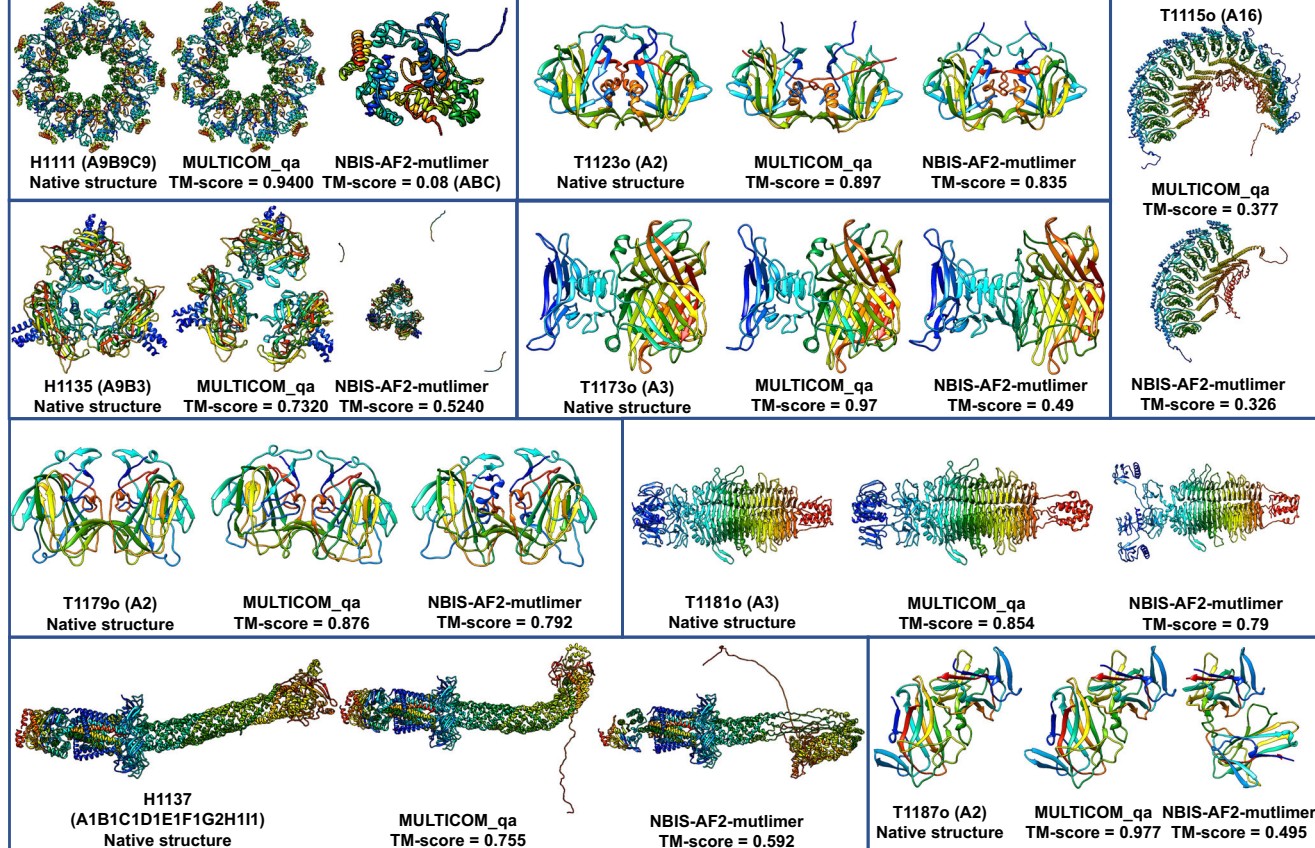

**Fig. 3 Nine notable cases where MULTICOM_qa outperformed NBIS-AF2-multimer.** The nine examples (H1111, T1187o, T1173o, T1115o, H1135, T1181o, T1123o, T1179o, H1137) on which MULTICOM_qa performed substantially better than NBIS-AF2-multimer.

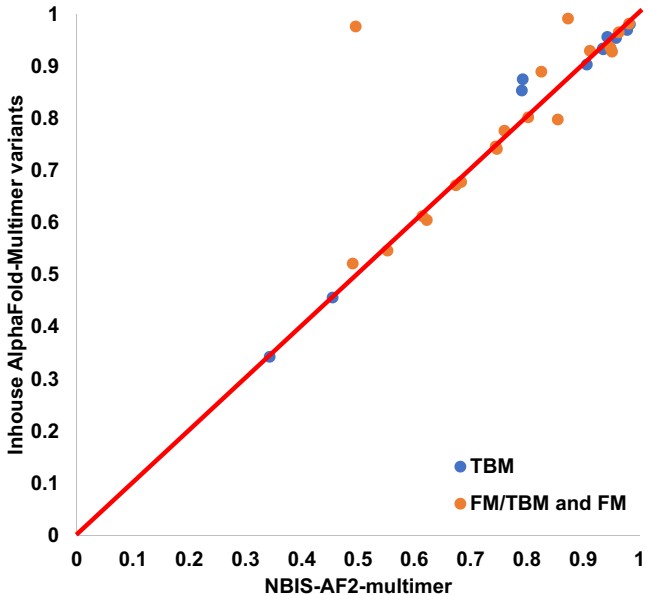

**Fig. 4 Comparison of MULTICOM predictions to NBIS-AF2-multimer on 31 multimer targets.** The TM-score of the best of top five predictions sampled from the diverse MSAs and templates generated by the sequence alignment component in the MULTICOM server system for each of 31 multimer targets (*y* axis) against that of NBIS-AF2-multimer (*x* axis).

length (>1536 residues), resolution (>4 Angstrom), and chain number (>8). The complexes were considered hetero-multimers based on a 0.9 sequence identity threshold between chains. Only hetero-multimers were used in this post-CASP15 experiment because all the 13 MSA generation methods were applied to them, while the homo-multimer structure prediction used only 7 kinds of MSAs without the complicated monomer MSA pairing. To remove artificial hetero-multimers due to the experimental artifact, we only retained hetero-multimers whose structure had at least ten inter-chain residue-residue pairs in contact with a minimum distance of < 5 Angstrom between any heavy atoms. To remove the hetero-multimers that are similar to known protein structures that may be used by AlphaFold-Multimer and MULTICOM, we excluded hetero-multimers whose subunits had more than 0.4 sequence identity with the monomer chains in the PDB prior to 04/01/2022. Additionally, a hetero-multimer was removed if any one of its subunits had a template hit in the monomer template database of MULTICOM consisting of monomer structures released by 04/01/2022 by HHSearch at the e-value threshold of 1. Moreover, the subunits of hetero-multimers were clustered using MMseqs2 with a 0.3 sequence identity threshold. The cluster ID assigned to each hetero-multimer was determined by the combination of the cluster IDs of its monomer chains. The hetero-multimer with the best resolution in each cluster was selected to be included into the final benchmark dataset. The dataset has 100 hetero-multimers in total.

For each of the hetero-multimers in the dataset, the $MSA_{paired}$ and the $MSA_{unpaired}$ generated by our in-house default AlphaFold-Multimer were also used to generate 25 predictions without using any templates. For a fair comparison, each of the 13 different kinds of paired MSAs of its subunits ($MSA_{paired}$) generated by MULTICOM together with the exactly same unpaired MSAs of the subunits ($MSA_{unpaired}$) was used by AlphaFold-Multimer to generate 25 predictions, without using any structural template as input. The top 5 predictions for each kind of $MSA_{paired}$ were selected by the AlphaFold-Multimer's

confidence score. The results of 13 kinds of MSA and the default AlphaFold-Multimer MSA are shown in Fig. S2, where each $MSA_{paired}$ is named by its interaction source and sequence database.

In Fig. S2a, the average TM-score of the top-1 predictions on the 100 hetero-multimers for the $MSA_{paired}$ generated by the default AlphaFold-Multimer (denote as default_multimer) is 0.799, which is higher than the average score of the other 13 kinds of $MSA_{paired}$ ranging from 0.7697 to 0.788. However, according to the one-sided Wilcoxon signed rank test, there is no significant difference between default_multimer and 7 kinds of $MSA_{paired}$ (spec_iter_uniprot_sto, str_iter_uniref_sto, str_iter_uniprot_sto, unidist_uniprot_sto, pdb_iter_uniref_sto, pdb_iter_uniref_a3m, and unidist_uniref_a3m). The average TM-score of the best of top 5 predictions of default_multimer is 0.8206, higher than the average score of the 13 MSAs ranging from 0.7954 to 0.8153, but there is no significant difference between default_multimer and 5 kinds of $MSA_{paired}$ (str_iter_uniref_sto, str_iter_uniprot_sto, spec_iter_uniprot_sto, pdb_iter_uniref_a3m, and str_iter_uniref_a3m) (Fig. S2b).

To investigate the effectiveness of combining the 13 kinds of $MSA_{paired}$, the default_multimer was employed to generate 325 predictions for each hetero-multimer, which were used to compare with the 325 predictions in the combined prediction pool of the 13 kinds of $MSA_{paired}$ (denote as combine). Notably, the average TM-score of the combine method for top 1 (or best of top 5) predictions selected by the AlphaFold-Multimer's confidence score is 0.8045 (or 0.8317), higher than the 0.7997 (or 0.8212) of default_multimer (Fig. S2c), even though the difference is not statistically significant. The results show that even though each of the 13 kinds of MSAs does not perform better than the default MSA, the predictions generated form them as whole have better quality than the default MSA, demonstrating their complementarity across different targets. Interestingly, only increasing the number of predictions from 25 to 325 for the default_multimer resulted in a much smaller improvement (e.g., 0.0004 TM-score difference for top 1 predictions and 0.0006 TM-score difference for top five predictions). The results show that sampling predictions with diverse MSAs can improve the quality of the assembly structure prediction more substantially than only increasing the number of predictions generated.

**Foldseek structure alignment-based multimer structure generation improves prediction accuracy.** During the CASP15 experiment, the Foldseek Structure Alignment-based Multimer structure Generation method (FSAMG) was applied to generate structural predictions for 26 multimers. For each multimer target, FSAMG was run 2–5 times with different tertiary structures predicted for the subunits/chains of the target, leading to 10–25 multimer predictions generated. On the 26 common targets, the average TM-score of the best of top five predictions ranked by AlphaFold-Multimer confidence score and generated by FSAMG is 0.81, higher than 0.79 of NBIS-AF2-multimer, showing that a noticeable improvement has been made by FSAMG over the standard sequence-alignment-based multimer structure generation in AlphaFold-Multimer.

Compared to NBIS-AF2-multimer, FSAMG generated much better predictions on H1140 (0.818 vs 0.622), H1144 (0.890 vs 0.683), T1173o (0.973 vs 0.490), and T1123o (0.893 vs 0.825) as shown in Fig. 5 and Fig. 6 due to several factors below, respectively.

For H1140 (stoichiometry: A1B1), a nanobody target, the MSA for each subunit/chain found by the sequence search (i.e., $MSA_{unpaired}$) was augmented by structural alignments generated by using Foldseek to search the tertiary structure of each chain against the known structures in the PDB. The augmented

$\text{MSA}_{\text{unpaired}}$ of each chain and the similar structural templates found by the Foldseek search were used as input for AlphaFold-Multimer to generate predictions, even though no paired alignments covering the two chains of H1140 were found by the Foldseek search. The highest TM-score of the predictions generated by FSAMG is 0.818 (Fig. 6), much higher than 0.622 of NBIS-AF2-multimer and 0.626 of our in-house AlphaFold-

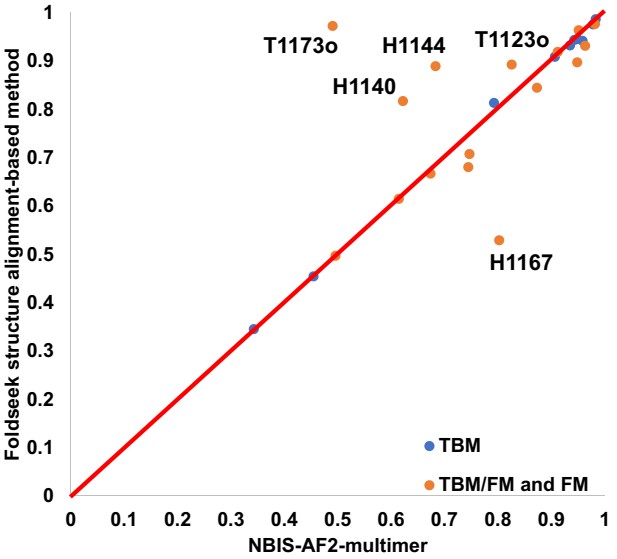

**Fig. 5 Comparison of Foldseek structure alignment-based multimer structure generation and NBIS-AF2-multimer predictions on 26 CASP15 multimer targets.** The TM-score of the best of top 5 predictions for each target generated by Foldseek structure alignment-based multimer structure generation (FSAMG) versus NBIS-AF2-multimer on 26 CASP15 multimer targets.

Multimer with the sequence alignment-based MSAs and templates as input. For H1144 (stoichiometry: A1B1), another nanobody target, the highest TM-score of the predictions generated by FSAMG is 0.890 (Fig. 6), higher than 0.683 of NBIS-AF2-multimer and 0.855 of our in-house AlphaFold-Multimer with the sequence alignment-based MSAs and templates as input. The multimer predictions generated by the two methods have very similar tertiary structures for individual chains. However, the multimer predictions generated by FSAMG have better interactions between the two chains than NBIS-AF2-multimer. Indeed, the main challenge for this target is to predict the interaction between the two subunits because there is no inter-chain co-evolutionary information in the MSAs of nanobody targets. For FSAMG, AlphaFold-Multimer was provided with the $\text{MSA}_{\text{unpaired}}$ containing newly added structural alignments as well as only two paired alignments in $\text{MSA}_{\text{paired}}$ to generate predictions. In contrast, for both H1140 and H1144, our other AlphaFold-Multimer variants that enabled MSA pairing cannot generate good predictions with ~1000 paired alignments in the $\text{MSA}_{\text{paired}}$. The results indicate using more paired MSAs with AlphaFold-Multimer results in bad predictions for nanobody targets because their two chains do not have co-evolution.

For T1123o (stoichiometry: A2), the number of sequences of the initial $\text{MSA}_{\text{paired}}$ was 26. FSAMG added 10 more structural alignments into the $\text{MSA}_{\text{paired}}$ and found 4 monomer templates (5W1NA, 5KOUA, 5W1ND, 5KOVA) for each subunit that were fed into AlphaFold-Multimer to generate 10 predictions. The top 5 predictions selected by the AlphaFold-Multimer confidence score have TM-scores of 0.873, 0.873, 0.877, 0.893 and 0.881 (Fig. 6), all higher than 0.825 of NBIS-AF2-multimer.

For T1173o (stoichiometry: A3), the number of sequences in the initial sequence alignment-based $\text{MSA}_{\text{paired}}$ was already larger than the maximum number of sequences (2048) that can be used by AlphaFold-Multimer, the paired alignments added into the $\text{MSA}_{\text{paired}}$ by FSAMG made little difference. The main difference

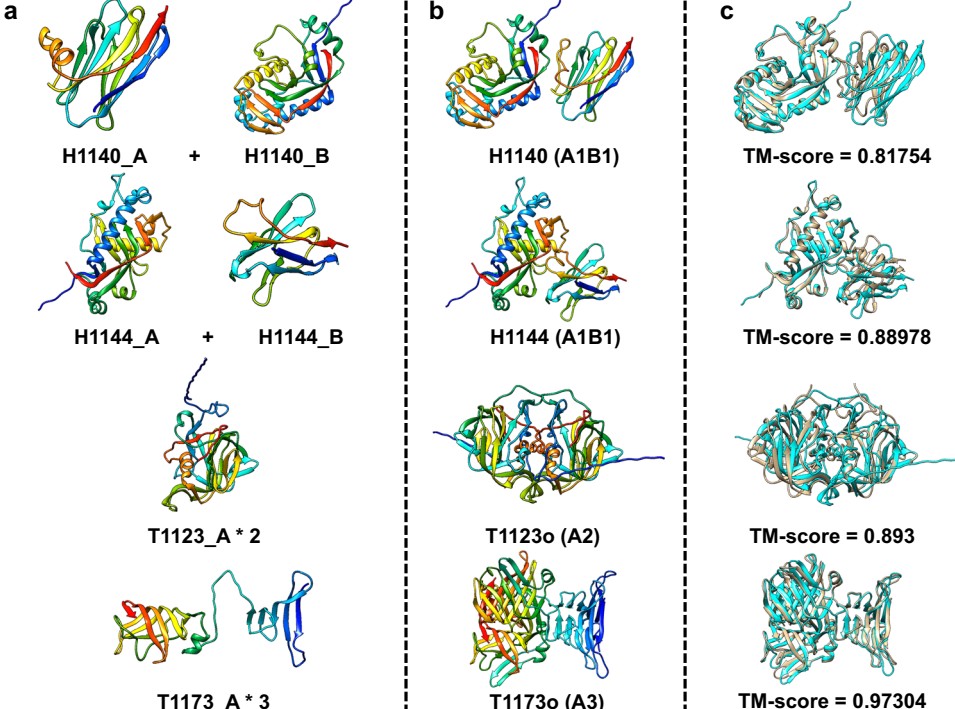

**Fig. 6 Four good predictions (H1140, H1144, T1123o, and T1173o) made by the Foldseek structure alignment-based multimer structure generation (FSAMG). a** Input monomer structures generated by AlphaFold2. **b** a multimer prediction generated by FSAMG. **c** The superposition between the multimer prediction and the native structure (cyan: prediction, gold: native structure) and the TM-score of the prediction.

is that FSAMG found 4 significant monomer templates including 4UW7A, 4UW4B, 4UW7C from a homo-trimer 4UW7, and 5AQ5B from another homo-trimer 5AQ5 for T1173o that were used as input for AlphaFold-Multimer to generate predictions. The proportion of high-accuracy predictions (TM-score > 0.95) generated by FSAMG is 60% (Fig. 6), while NBIS-AF2-multimer and our other AlphaFold-Multimer variants did not generate any prediction of such high accuracy.

Among 26 multimer targets, FSAMG performed only obviously worse than NBIS-AF2-multimer on H1167—an antibody-antigen target. The best TM-score of its top 5 predictions is only 0.529, much lower than 0.802 of NBIS-AF2-multimer. The reason is that there were two kinds of conformations (a bad one with TM-score $\simeq$ 0.5 and a good one TM-score $\simeq$ 0.8) in the prediction pool generated by FSAMG for H1167. The top 5 predictions selected by the confidence score unfortunately all belong to the bad conformation.

**The comparison of the multimer structure QA methods**. In the CASP15 experiment, the three main QA methods, including the AlphaFold-Multimer self-reported confidence score (Confidence), the average pairwise similarity score between a prediction and all other predictions for a target (PSS) calculated by MM-align, and the average of the two scores (CoPSS), were applied to rank and select multimer predictions by MULTICOM predictors.

We use the average per-target ranking loss and average per-target correlation to compare the three QA methods on all the full-length predictions generated for 31 multimers by the CASP15 server prediction deadline (called server_prediction_dataset) and by the CASP15 human prediction deadline (called human_prediction_dataset). server_prediction_dataset is a subset of human_prediction_dataset. For some multimer targets, human_prediction_dataset includes some additional predictions generated between the server prediction deadline and the human prediction deadline. The per-target ranking loss for a target is the difference between the TM-score/DockQ score of the best prediction for the target in a dataset and the TM-score/DockQ score of the no. 1 prediction selected for the target by a QA method. The smaller the loss, the better the ranking for the target. The per-target loss is averaged over all the targets to assess the ranking performance of a QA method. The per-target correlation for a target is Pearson's correlation between the quality scores generated by a QA method for the predictions and the true quality scores (TM-scores/DockQ score) of the predictions. Higher the per-target correlation, the better the quality scores generated by the QA method. The per-target correlation can be averaged over all the targets to assess the prediction accuracy estimation capability of a QA method.

The average per-target ranking loss and average per-target correlation of the three QA methods on the two datasets are reported in Table 2. On the server_prediction_dataset, CoPSS has the lowest average ranking loss and highest average correlation of 0.0842 and 0.3898, better than 0.0866 0.3447 of Confidence and

0.0853 and 0.3767 of PSS in terms of TM-score, indicating that combining Confidence and PSS improves the performance of estimating the global accuracy of the predictions in the server_prediction_dataset. On some targets, PSS can substantially outperform AlphaFold-Multimer's confidence score. For instance, PSS's TM-score ranking loss for T1179o is 0.03, much lower than 0.428 of AlphaFold-Multimer's confidence score. In terms of DockQ score, Confidence has the lowest average ranking loss on the server_prediction_dataset (0.1003), while CoPSS has the highest average correlation on the server_prediction_dataset (0.4073), indicating that they have some complementarity.

On the human_prediction_dataset, in terms of TM-score, Confidence yields the lowest loss of 0.0505 but the lowest correlation of 0.3845, while CoPSS has the second lowest loss of 0.0625 and the highest correlation of 0.4078, indicating that combining Confidence and PSS as CoPSS achieves better performance than PSS in terms of global structural accuracy (TM-score). For DockQ score that more specifically considers the quality of the interface in predicted structures, Confidence has the lowest average ranking loss (0.0979), while CoPSS has the highest correlation (0.459). Based on the results of the two datasets, Confidence, and PSS are complementary for estimating the accuracy of multimer predictions, while Confidence performs better in estimating the interface accuracy of the multimer predictions in terms of the loss of DockQ score. Combining them may be useful to improve the QA of multimer predictions. However, how to combine them to achieve consistently better results still needs more investigation.

**The performance of Foldseek structure alignment-based multimer structure refinement**. The Foldseek structure alignment-based multimer structure refinement method (FSAMR) was applied to 19 multimer targets during the CASP15 experiment. The per-target average maximum TM-scores of the original predictions is 0.752, similar to 0.750 of the refined predictions. However, FSAMR was able to generate better predictions for some targets, especially for T1187o (TM-score 0.899 vs 0.689) (Fig. S3). For T1187o, the improvement may be due to the extra alignments added to the $MSA_{paired}$ by FSAMR. However, FSAMR can also generate predictions of worse quality. One extreme case is T1153o (Fig. S3), where a refined prediction has a TM-score of 0.484, much lower than 0.928 of the initial prediction. However, the worse quality can be detected by the change in the AlphaFold-Multimer confidence score of the predictions. The confidence scores for the 5 original predictions are close to 0.9, while the confidence scores for the five refined predictions are close to 0.48, indicating a significant drop in the confidence score after the refinement. If we only use the refined predictions whose confidence score is higher than that of the initial prediction by at least a margin (i.e., 0.2), the average per-target maximum TM-scores of the refined predictions is 0.752, higher than 0.740 of the initial predictions. The results show that FSAMR can be used to

**Table 2 The average per-target ranking loss and average per-target correlation of the three QA methods (Confidence, PSS, and CoPSS) on the server_prediction_dataset and the human_prediction_dataset.**

| QA Method | server_prediction_dataset | | | | human_prediction_dataset | | | |
|---|---|---|---|---|---|---|---|---|
| | TM-score | | DockQ score | | TM-score | | DockQ score | |
| | Loss↓ | Corr↑ | Loss↓ | Corr↑ | Loss↓ | Corr↑ | Loss↓ | Corr↑ |
| Confidence | 0.0866 | 0.3447 | **0.1003** | 0.3776 | **0.0505** | 0.3845 | **0.0979** | 0.4328 |
| PSS | 0.0853 | 0.3767 | 0.1373 | 0.3455 | 0.0892 | 0.3853 | 0.1224 | 0.4257 |
| CoPSS | **0.0842** | **0.3898** | 0.1053 | **0.4073** | 0.0625 | **0.4078** | 0.1171 | **0.4590** |

The bold font highlights the best result. The underline denotes the second-best result.

generate some diverse and even better predictions for a multimer target if the change in the prediction confidence score is substantial.

**MULTICOM server assembly predictors versus human assembly predictors.** Compared to the MULTICOM server predictors, the prediction pool of the MULTICOM human predictors (MULTICOM and MULTICOM_human) was slightly larger since some additional predictions for some hard targets were generated by either the customized AlphaFold-Multimer with different inputs or by FSAMR between the server prediction deadline and the human prediction deadline. The average TM-score of the best of five predictions for 41 multimer targets by MULTICOM_qa is 0.796, only slightly lower than 0.797 of the best MULTICOM human predictor (MULTICOM_human), indicating that they achieved largely comparable performance. However, the average TM-score of the top 1 predictions for the 41 multimer targets of MULTICOM_qa is 0.757, lower than 0.776 for MULTICOM_human. The improvement made by the human predictor comes mostly from the increase in the number of multimer predictions generated for some targets and some extra human-guided prediction ranking and combination, especially on the top 1 prediction. For instance, for T1174o and T1181o, there were two alternative conformations in the top 5 predictions submitted by MULTICOM_qa, but it used the bad conformation as the top 1 prediction, while MULTICOM_human used the good conformation as the top 1 prediction. For a large hard target T1176o, more structural predictions were generated by MULTICOM_human for the components of T1176o to be combined to generate full-length predictions for T1176o. MULTICOM_human's best prediction has a TM-score of 0.249, higher than 0.196 of the best prediction by MULTICOM_qa.

**Relationship between MSA and multimer structure quality.** For tertiary structure prediction, the quality of the input MSA quantified by the number of effective sequence (Neff) was shown to have a high correlation coefficient (i.e., 0.777) with the quality score (i.e., GDT-TS) of the tertiary structure predictions generated by AlphaFold2 for the single-chain monomer targets in CASP15[22]. However, it is more difficult to study the relationship between the quality of MSA and the quality of multimer structural predictions because AlphaFold-Multimer takes both the MSA of individual chains ($MSA_{unpaired}$) and the paired MSA of the multimer ($MSA_{paired}$) as input, while AlphaFold2 only uses one MSA as input for tertiary structure prediction. Specifically, for homo-multimer consisting of multiple identical chains, AlphaFold-Multimer uses only $MSA_{paired}$ as input, but for hetero-multimer, AlphaFold-Multimer leverages both $MSA_{unpaired}$ and $MSA_{paired}$ if available. Here, we use the results of our default AlphaFold-Multimer variant (default_multimer in Table S2 for homo-multimer and in Table S3 for hetero-multimer) in the MULTICOM system to study the relationship between MSA quality and prediction quality. The quality of the input MSAs for a multimer is calculated by $Neff_{multimer} = \sum_{i=1}^{n} \frac{L_i}{L} Neff(MSA_i)$, where $n$ is the number of subunits of the multimer, $MSA_i$ is the combination of $MSA_{unpaired}$ for subunit $i$ and the portion of the alignment in $MSA_{paired}$ for subunit $i$, $L_i$ is the sequence length of subunit $i$, $L$ is the total sequence length of the multimer. The per-target average correlation between the average TM-scores of the predictions and $Neff_{multimer}$ of the MSAs is 0.240 on 31 multimer targets, which is a much weaker correlation than the tertiary structure prediction for single-chain monomer targets. The weak correlation may be because the quality of multimers depends not only on the quality of MSAs of individual chains but also on the

quality of the MSAs informing the interaction between the chains. But this quality of MSAs is not well measured by $Neff_{multimer}$.

**Prediction of the structures of very large assemblies.** Several multimer targets (e.g., H1111, H1114, H1137, and T1115o) are so large that AlphaFold-Multimer could not generate full-length predictions for them directly because the 80GB memory of the Nvidia A100 GPU used by MULTICOM was not sufficient to handle them. In this situation, MULTICOM decomposed each of such targets into multiple components to predict the structures of components separately and then combined the structural predictions of the components into the full-length of the target through the overlapped chains between the components. For instance, H1137 (stoichiometry: A1B1C1D1E1F1G2H1I1) has 9 different chains and 3,939 residues in total. Based on the structure template information, the first domains of six chains (A1B1C1D1E1F1) form a ring, and the ring structure interacts with H and I Chains. Therefore, MULTICOM first predicted the structure of the six chains (A1B1C1D1E1F1) (see Fig. S4a) for two typical conformations predicted for them: a ring with the straight tail and a ring with the bended tail). It then divided the six chains into a ring structure and a tail structure. The sequences of the ring structure of the first six chains were then cut off to be used with the other three chains (G, H, I) to predict the structure of the 9 chains excluding the tail of the first six chains (see Fig. S4b). Finally, the structure of the first six chains and the structure of the 9 chains without the tail were combined by Modeller[23] through their common ring structure to build the full-length structure for H1137 (see Fig. S4c for the bended conformation predictions for H1137 and their TM-score as well as the native structure of H1137). The full-length structure with a straight tail (see Fig. S4a) has better quality than the one with the bended tail, but the latter is more frequent than the former. The AlphaFold-Multimer confidence score could have selected the structure with the straight tail correctly, but the PSS score preferred the inferior structure with the bended tail because it was more abundant.

It is worth noting that AlphaFold-Multimer was rather effective in generating structural predictions for small complexes with diverse sampling strategies in CASP15. For the small complexes (e.g., less than 6 chains), extensive sampling approaches (such as generating over 1000 predictions) through AlphaFold-Multimer employed by some groups such as the Wallner group yielded some high-quality structural predictions. The main challenge for these approaches lies in selecting the structural prediction with the highest quality, which can be tackled by developing more effective methods for assessing the quality of multimer structures.

However, the prediction complexity intensifies when dealing with large higher-order complexes due to two main factors. Firstly, predicting the multimer structure for higher-order complexes demands substantial computational resources (e.g., H1111, H1114, T1115o, H1137) that may not be available. In this case, dividing a large multimer into subcomplexes to generate predictions for them to be combined into full-length predictions for the multimer is a viable option. However, a large multimer usually has too many sub-complexes to generate predictions for in a limited amount of time. Identifying critical sub-complexes that can link sub-complexes together to form the structure of the entire multimer is critical and sometimes very challenging. For instance, constructing full-length structures for H1111 (A9B9C9) and H1114 (A4B8C8) hinges on generating the structures of subcomplex C9 of H1111 and subcomplex A4 of H1114 because they form the backbone of linking all the chains in the multimers together. However, it is not always obvious to prioritize them for

structure generation. Secondly, higher-order complexes may have multiple protein-protein interaction interfaces between their units, which presents a challenge for AlphaFold-Multimer to predict and combine them to generate full-length multimer structures due to a large number of combinatorial choices. For example, there are more than four possible interaction interfaces between two identical subunits of T1176o (A8) predicted by AlphaFold-Multimer. No full-length prediction for this target submitted from the CASP15 predictors has a TM-score higher than 0.5. Addressing these challenges may require the development of efficient and effective methods for the combinatorial problems of selecting subcomplexes to generate predictions and combining different predictions with different interfaces to generate full-length structures of multimers.

## Conclusions

We report a protein prediction system (MULTICOM) to improve AlphaFold-Multimer-based complex structure prediction, which blindly participated in the CASP15 experiment from May to August 2022 as both server and human predictors. MULTICOM enhances AlphaFold-Multimer predictions by generating diverse MSAs and structural templates using both sequence and structure alignments for AlphaFold-Multimer to generate better predictions, combining AlphaFold-Multimer confidence score with the complementary pairwise prediction similarity score to rank predictions, and further refining the predictions using Foldseek structure alignment to augment MSAs and templates input for AlphaFold-Multimer. MULTICOM_qa server ranked among the top CASP15 server predictors for assembly structure prediction and performed significantly better than a standard AlphaFold-Multimer predictor. The results show that using diverse MSAs and structural templates as input is an effective way to generate better predictions for assembly structure prediction. Particularly, the FSAMG method performs better than the existing sequence alignment-based approach used by AlphaFold-Multimer. Moreover, the FSAMR can substantially improve the quality of structural predictions for some targets. Furthermore, our results show that the average pairwise similarity between a prediction and other predictions is complementary with AlphaFold-Multimer's self-reported confidence score for estimating the accuracy of assembly predictions.

## Methods

**The MULTICOM protein complex structure prediction system and methods**. The workflow of the MULTICOM complex/multimer prediction system consists of seven steps (Fig. 7): (1) single-chain (monomer) structure prediction for each unit of a multimer, (2) monomer MSAs concatenation, (3) monomer templates concatenation, (4) multimer structure generation, (5) Foldseek structure alignment-based multimer structure generation, (6) multimer structural prediction ranking, and (7) Foldseek structure alignment-based multimer structure refinement. The method in each step is described as follows.

**Single-chain structure prediction for each subunit of a multimer**. Our in-house single-chain (monomer) tertiary structure prediction system[22] built on top of AlphaFold v2.2.0 is used to generate MSAs, structural templates, and predicted tertiary structures for each subunit of a multimer target. It uses sequence alignment tools including HHblits[24], JackHMMER[25], MMseq2[26], an in-house implementation of DeepMSA[27] to search multiple protein sequence databases including UniClust30[28] (uniclust30_2018_08), UniRef30[28] (UniRef30_2021_02), Uniref90[29] (version 04/24/2022), UniProt[29] (version 04/24/2022), the Integrated Microbial Genomes (IMG) database[30] and the

metagenome sequence databases (e.g., BFD[31,32], Metaclust[32], MGnify clusters[33]) to generate a diverse set of MSAs for each unit (monomer).

**Monomer MSAs concatenation**. AlphaFold-Multimer uses two kinds of MSAs as input: (1) the unpaired MSA for each subunit ($MSA_{unpaired}$) and (2) the paired MSA that may encode the coevolutionary information between the subunits ($MSA_{paired}$), which are prepared as follows by the MULTICOM system.

For hetero-multimers, the alignments in the MSAs of the subunits are concatenated using the potential protein-protein interaction information extracted from multiple sources to construct $MSA_{paired}$ as shown in Table S4, including species annotations, UniProt accession number of sequences, protein-protein interactions in the STRING database[34] and the complex structures in the Protein Data Bank[35] (PDB). The alignment description (header) in UniClust30, UniRef30, UniRef90, and UniProt contains the UniProt ID, UniProt accession number, and the species annotations (e.g., Organism identifier (OX)[13], Organism name (OS)[36], Taxonomy identifier (Tax)[36]). Based on the species information, the individual sequence alignments in the MSAs of the subunits belonging to the same species are concatenated to generate the paired multimer sequence alignments sequentially in a top-down manner. Based on UniProt accession numbers, sequence alignments from the subunit MSAs are concatenated if the difference between their UniProt accession numbers is smaller than 10 as in RosettaFold[37]. For simplification, the alignments with the same UniProt accession number prefix (e.g., except for the last character) are paired. The STRING database (version v11.0) contains many hypothetical protein-protein interactions, each of which has an interaction score. The interaction score between two protein sequences in the UniProt database is retrieved according to the mapping between STRING ID to the UniProt ID. Two sequence alignments from two subunit MSAs are concatenated if their interaction score is higher than 0.5. According to the mapping between the PDB code and UniProt ID, two sequence alignments from two subunit MSAs are concatenated if they are mapped to the same PDB code indicating that they are two subunits of the same protein complex. The four sources of potential protein-protein interactions above are used by MULTICOM to generate 13 kinds of $MSA_{paired}$ for hetero-multimers from the different databases (Table S4). The $MSA_{unpaired}$ for hetero-multimers is always generated by the same default MSA generation procedure in AlphaFold-Multimer (e.g., searching the subunit/chain sequence against UniRef30 and BFD, and MGnify clusters to generate MSAs). Both $MSA_{paired}$ and $MSA_{unpaired}$ are used in the multimer structure generation for hetero-multimers by AlphaFold-Multimer.

For homo-multimers, the default MSA generation of the AlphaFold-Multimer is used by MULTICOM on different sequence databases to generate several kinds of $MSA_{paired}$ (see default_multimer, default_pdb, default_pdb70, default_comp, default_struct, default_af, and default_img in Table S2) where the $MSA_{unpaired}$ is simply concatenated horizontally together as the $MSA_{paired}$ since the $MSA_{unpaired}$ of each subunit is identical. In contrast, the customized multimer MSA generation methods in Table S2 pair only the alignments in the MSAs of the subunits of the multimer that have the same species annotation or PDB complex codes to generate $MSA_{paired}$. The alignments in the MSAs of the subunits without the species annotation or whose UniProt IDs cannot be mapped to any PDB codes are paired with gaps. Only $MSA_{paired}$ is used in the multimer structure generation for homo-multimers by AlphaFold-Multimer, while $MSA_{unpaired}$ is ignored.

**Monomer templates concatenation**. The sequences of the subunits in the multimer are searched against the publicly available pdb_seqres database (version 04/24/2022), pdb70 (version 03/13/2022) monomer

## MULTICOM  Protein complex structure prediction system

**Fig. 7 The workflow of the MULTICOM protein complex structure prediction system.** Its prediction process starts with the prediction of the tertiary structure of each chain (monomer) of a multimer target (Box 1). Then, on one hand, the MSAs and templates of the monomers are concatenated together (Box 2 and Box 3) as input for AlphaFold-Multimer to generate multimer structure predictions (Box 4). On the other hand, the predicted tertiary structures of the monomers are used by the Foldseek structure alignment-based multimer structure generation method to generate multimer structure predictions (Box 5). The multimer structure predictions generated by the two approaches above are pooled together to be evaluated and ranked (Box 6). Finally, the top-ranked models are refined by the Foldseek structure alignment-based multimer structure refinement method (Box 7).

template database[16] curated from Protein Data Bank (PDB), an in-house monomer template database pdb_sort90[22], and an in-house protein template database (pdb_complex) constructed from only the biological assemblies in the PDB using HHSearch[38], resulting four kinds of templates. pdb_complex was constructed in a similar way as pdb_sort90 except that the former only considered the biological assemblies in the PDB while pdb_sort90 considered all the proteins in the PDB. The templates found for each subunit are concatenated together if they share the same PDB code. Only one concatenated multimer template is kept for each PDB code. Finally, the predicted tertiary structures for each subunit/chain are also used as the fifth kind of templates, which can lead to highly inflated AlphaFold-Multimer confidence scores for generated predictions and is less useful than the other four kinds of templates.

**Multimer structure generation**. A customized version of AlphaFold-Multimer v2.2.0 that accepts pre-generated MSAs (e.g., $MSA_{unpaired}$ and $MSA_{paired}$) and structural templates above as input is used to generate predictions. To perform more extensive sampling, the value of parameter num_ensemble_eval is changed from 1 to 3 and num_recycle from 3 to 5 in the customized AlphaFold-Multimer. The customized AlphaFold-Multimer takes up to 19 combinations of MSAs and structural templates (Table S2) for homo-multimer and up to 29 combinations of MSAs and structural templates (Table S3) for hetero-multimer as input to generate 10 structural predictions for each combination by setting the value of num_multimer_predictions_per_model to 2. Only the top 5 predictions ranked by the AlphaFold-Multimer confidence scores for each MSA-template combination are added into the structural prediction pool for

the multimer, resulting in up to 95 (or 145) predictions generated for each homo-multimer (or hetero-multimer) target.

**Foldseek structure alignment-based multimer structure generation**. Different from using the sequence alignment-generated MSAs and templates above as input for AlphaFold-Multimer to generate predictions, we developed a FSAMG method (Fig. S5) to generate up to 25 predictions as follows. The predicted tertiary structures of the subunits of a multimer generated by AlphaFold2 are searched against both the pdb_complex template database and the tertiary structure predictions in the AlphaFoldDB (the version 1 released before March 2022) by a fast structure alignment tool—Foldseek—to identity similar structural hits. The output of the Foldseek search includes the e-value of the structural hits as well as the structural alignments between the target prediction and the hits. The structural alignments of local structural hits with $e$ value $\leq 0.001$ and global structural hits with TM-score $\geq 0.3$ are converted into the sequence alignments between the target and the hits.

For hetero-multimers (Fig. S5a), the $MSA_{unpaired}$ is initialized as the MSA generated for each subunit by the tertiary structure prediction system, while the $MSA_{paired}$ is set empty. Two sequence alignments generated from structural alignments for two subunits/chains of the hetero-multimer are paired if they come from the same PDB protein complex (i.e., sharing the same PDB code but from different chains) or from the two non-overlapping domains of a hit in the AlphaFoldDB. Only one paired alignment is kept for each PDB code to avoid redundancy in the paired alignments. The sequence alignments are added into the $MSA_{unpaired}$ for each subunit, while the paired sequence alignments are included into the $MSA_{paired}$.

For homo-multimers (Fig. S5b), two different approaches are applied to generate the $MSA_{paired}$. The first approach initializes the $MSA_{paired}$ with the horizontal concatenation of the multiple copies of the MSA of monomer generated by the tertiary structure prediction system. Subsequently, the sequence alignments of individual subunits/chains that share the same PDB code but from different chains or from the non-overlapping domains of a hit in the AlphaFoldDB are paired. The top 50 paired alignments ranked by e-value and TM-score are added into the $MSA_{paired}$. The second approach initializes the $MSA_{paired}$ by the pairings of sequence alignments generated from the structural alignments that share the same PDB code or ID in the AlphaFoldDB. Then the horizontal concatenations of the multiple copies of the MSA of monomer generated by the tertiary structure prediction system are added to the $MSA_{paired}$.

For each subunit in the hetero-multimer and homo-multimer, the similar structural hits from the pdb_complex template database and the AlphaFoldDB are ranked by e value and TM-score. The ranked structural hits are treated as monomer templates for each subunit. The structure-alignment generated MSAs and the monomer templates for each subunit are used as input for the customized AlphaFold-Multimer to generate 10 predictions. The top 5 predictions ranked by their confidence scores are added to the structural prediction pool for the multimer. This procedure is applied with 2–5 top-ranked tertiary structure predictions of the subunits of the multimer as described above to generate 10 to 25 predictions in total. This structure alignment-based method can find some similar structural hits for hard targets that sequence alignment methods cannot, leading to deeper MSAs and more structural templates, which can be used by AlphaFold-Multimer to generate better structure prediction.

**Multimer structural prediction ranking**. MULTICOM applies three QA methods to rank the multimer predictions. Firstly, the average pairwise structural similarity (PSS) score between a prediction and other predictions in the prediction pool of a multimer is used to rank the structural predictions[39]. The pairwise structural similarity score is calculated by MM-align[40]. Secondly, the confidence score generated by AlphaFold-Multimer for each prediction is also used to rank the predictions. Finally, the average of the two is applied to rank the predictions.

**Foldseek structure alignment-based multimer structure refinement**. Given an initial multimer prediction and its MSAs (i.e., $MSA_{unpaired}$ and/or $MSA_{paired}$), the tertiary structure of each subunit in the multimer structural prediction is used as input for Foldseek to search for similar structures in the pdb_complex template database and the AlphaFoldDB (the version 1 released before March 2022). The structure alignments with e value $\leq 0.001$ or TM-score $\geq 0.3$ between each subunit and structural hits are converted into sequence alignments. The sequence alignments of the subunits generated from the Foldseek search are concatenated if they are from the same PDB complex structure or the non-overlapped regions of the same single-chain AlphaFoldDB prediction to construct the MSA for the multimer. The top structural hits of the subunits are used as the monomer templates for each subunit of the multimer. The concatenated MSAs are ranked by e-value and TM-score. Only the top 50 concatenated MSAs are added to the original $MSA_{paired}$ to generate a deeper MSA. The augmented $MSA_{paired}$, original $MSA_{unpaired}$ (if any for hetero-multimers), and the templates are used as inputs for the customized AlphaFold-Multimer to generate the refined predictions. If the highest confidence score of the newly refined predictions is higher than that of the input prediction, the refinement process is repeated with the refined prediction and its MSAs as input until the number of refinement iterations reaches 5. The refined prediction with the

highest confidence score generated in the refinement process is used as the final output prediction.

**Implementation of the CASP15 assembly structure predictors**. During CASP15, the MULTICOM protein assembly structure prediction system was mainly executed on three computer servers (server 1: 192 AMD EPYC 7552 48-Core CPU Processor, 377 GB RAM, an NVIDIA A100 PCIe 80GB GPU; server 2: 192 AMD EPYC 7552 48-Core CPU Processor, 503 GB RAM, an NVIDIA A100 PCIe 40GB GPU; and server 3: 192 AMD EPYC 7552 48-Core CPU Processor, 1 TB RAM, an NVIDIA A100 PCIe 40GB GPU) respectively to generate the predictions for multimer targets before the server prediction deadline and additional predictions for some multimer targets between the server prediction deadline and the human prediction deadline if necessary. Generally, ~15–195 predictions were generated for each target, depending on its size. The two CASP15 multimer server predictors (MULTICOM_qa and MULTICOM_deep) mainly used the AlphaFold-Multimer confidence score and the average of the confidence score and the PSS score to rank multimer predictions, respectively.

The two human multimer predictors (MULTICOM and MULTICOM_human) considered all the predictions generated before the human prediction deadline. Moreover, the Foldseek structure alignment-based multimer structure refinement was applied to refine the top-ranked predictions of most targets, and the refined predictions were added to the prediction pool for the final multimer structural prediction ranking. Generally, about 40–315 predictions were generated for each human target. MULTICOM_human mainly used the average of the confidence score and the PSS score to rank and select predictions for final submission, while MULTICOM mainly applied the PSS score to rank predictions. The ranking may be manually adjusted according to human inspection. The main difference between the MULTICOM server and human predictors is summarized in Table S5.

For some very large complexes (e.g., H1111, H1114, H1135, H1137, T1115o, T1176o, and T1192o), no full-length multimer predictions or only poor full-length predictions could be generated by AlphaFold-Multimer due to the GPU memory limitation, the template-based structure modeling based on Modeller[23] was applied to combine the predictions of the components of the complexes generated by AlphaFold-Multimer.

**Statistics and reproducibility**. 41 multimer targets were used in CASP15, which is the largest blind multimer test dataset to date. The quality of the structural predictions submitted by the predictors is not normally distributed. Consequently, the non-parametric one-sided Wilcoxon signed rank test is the statistical test used when comparing the performance between different predictors at the 0.95 confidence level.

**Reporting summary**. Further information on research design is available in the Nature Portfolio Reporting Summary linked to this article.

## Data availability
The CASP15 data including the experimental structures are available at: https://predictioncenter.org/casp15/index.cgi. The protein structures predicted by the inhouse MULTICOM3 software are available at https://github.com/BioinfoMachineLearning/MULTICOM3/tree/main/evaluation. The source data of the Figures and Tables in this study can be obtained from Supplementary Data 1.

## Code availability
The source code of MULTICOM is available as an add-on package for AlphaFold-Multimer at: https://github.com/BioinfoMachineLearning/MULTICOM3 and Zenodo[41].

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

## Acknowledgements
We thank CASP15 organizers and assessors for making the CASP15 data available. We also thank Dr. Soeding's group for releasing the latest HHsuite protein hidden Markov model (HMM) database for the community to use prior to the CASP15 experiment. This work is partially supported by two NIH grants [R01GM093123 and R01GM146340], Department of Energy grants [DE-SC0020400 and DE-SC0021303], and three NSF grants [DBI1759934, DBI2308699, and IIS1763246].

## Author contributions

J.C. conceived the project. J.C. and J.L. designed the experiment. J.L., J.C., Z.G., T.W., R.R., F.Q., and C.C. performed the experiment and collected the data. J.L. and J.C. analyzed the data. J.L. and J.C. wrote the manuscript.

## Competing interests

The authors declare no competing interests.
