## [Peer Review File · Communications Biology]

Reviewers' comments:

Reviewer #1 (Remarks to the Author):

The authors have proposed a new protein prediction tool called MULTICOM3, based on AlphaFold-Multimer. In this work, the authors utilized various methods such as FoldSeek and deepmsa to generate structural templates and MSA (multiple sequence alignment) to enhance AlphaFold-Multimer. This method demonstrated good performance in CASP15.

1. How significant is the role of Pair-MSA in MULTICOM3? Due to the relatively small number of target proteins in CASP15, are there any comparative results with a benchmark dataset available?
2. The authors mentioned using various Pair-MSAs, but it is not clear how they were generated. The open-source code provided on GitHub was unable to successfully run and generate Pair-MSA for the test cases provided by the authors.
3. Regarding the results obtained from FoldSeek, are all the results used to enhance MSA, or are there specific strategies to select the good results for enhancing MSA? If there are selection strategies, please provide details.
4. How do you convert a structural alignment between two proteins into a sequence alignment?
5. For homomeric complexes, if the results obtained from FoldSeek for subunits do not meet the condition "Paired by PDB code or non-overlapping regions of AlphaFoldDB hit" as shown in Figure S3, does the FSAMG method only provide structural templates for individual subunits without the enhanced MSA component? If that is the case, can the performance of multimers still be improved in such situations?

Reviewer #2 (Remarks to the Author):

Jian Liu et al present a tool for quaternary structure prediction called MULTICOM, which is an improved version of the famous AlphaFold-Multimer prediction. In particular, they enhanced the MSA generation to provide to AlphaFold-Multimer, used a new complementary metric for ranking predictions and added a new refinement method based on Foldseek. The improvements in the quality of the predictions don't look spectacular with about 8% higher than AlphaFold on the TM-score, which means in general, the folding and interface should be conserved as shown in the CAPRI evaluation of the CASP15 round. However, the authors give here a fair presentation of the results, showing when their tool really improve the results and when it doesn't, with not exaggerated improvement. In addition, with Alphafold-Multimer, the predictions are at a high level because the tool with default parameters already provide very nice predictions for many targets. Therefore, improving its predictions is challenging and any enhancement is always an added-value for the scientific community, knowing moreover that MULTICOM very rarely degrades the predictions while, in the majority of cases, don't change or improve them. The authors also provide the tool on a github repository for the last version of AlphaFold-Multimer with details for installing and running it. I would recommend the publication but suggest a few points to be addressed to improve the manuscript and some bugs have to be fixed on the Github repository.

1) The beginning of the introduction (second paragraph) could be improved a bit about classical historical docking methods. "the accuracy of the protein docking is generally low" could be put into perspective with cases where these methods work well and cases where they fail. The references 1 and 2 that are given are not the most appropriate because they refer to protein-ligand methods while the article is only based on protein complex prediction. These references should be replaced by more appropriate and more recent ones.

2) "Results and conclusion" section should be renamed "Results and discussion" and although the sections are well explained, they could sometimes be put a bit more into perspective about for instance:

- what are the cases that still fail for AlphaFold with suggestions to improve these cases
- the effect of the extended sampling because with MULTICOM, the number of generated predictions is higher than the 25 predictions of AlphaFold-Multimer by default

3) Do the authors know which combination of MSA work the best ? If they have access to such an evaluation, they could precise it in the text.

4) In "The comparison of the multimer model quality assessment methods" section, the AlphaFold-Multimer confidence score is used. Because this score is based for 80% on the ipTM, I am curious to see what the loss and correlation would be comparing to the DockQ that is more focused on the interface, rather than the TM-score.

5) The term "model" is a bit vague, especially now that we talk about neural network models in structural bioinformatics. I would suggest to use the term "prediction" to avoid any confusion with any other type of model.

6) The version of AlphaFold-Multimer that was used for the article should be precised in the "Materials and Methods" section

7) Figure 1 for T1174o: are the authors sure about the TM-scores ? Because the structures look similar but the TM-scores are very different. If this is correct, could they explain why such a difference ?

8) Part "Overall performance of MULTICOM_qa compared with the standard AlphaFold-Multimer": that would be nice to show the structures H1114, T1176o, T1115o, T1160o and T1161o like in Figure 1 in Supplementary materials for instance

9) For FASMG, why 26 complexes and not more ?

10) In "Materials and Methods" section: String interaction scores are between 0 and 1, then could the authors explain how the 500 threshold was obtained ?

11) The differences between all the MULTICOM_xxx versions are not always clear. I suggest a table that explains the differences for each version in Supplementary Materials

12) What are the differences between the Github version and the one used in the paper ? I see that the last AlphaFold version (2.3.2) that the user installs can be used, which is very good, but are there any other differences because the article says that a customized version of AlphaFold was used ?

13) What is the best configuration (memory/CPU) to run the tool ? That could be useful to add it in the Materials and Methods section so that the user knows for its installation.

Minor points:

14) Sometimes the term "AlphaFold-Multimer" is used and sometimes "AlphaFold2-Multimer" is used, one of both should be chosen and used everywhere. In addition, please check the spelling (e.g.: AlphaFold-Multimer)

15) CASP is "Critical Assessment of protein Structure Prediction"

16) In "Overall performance of MULTICOM_qa compared with the standard AlphaFold-Multimer" section: "(H1111, T1187o, T1173o, T1115o, H1135, T1181o, T1123o, T1179o, H1137)" is repeated three times. Please avoid repetitions in this part.

17) "For instance, for T1174o and T1181o, there were two alternative conformations in the top 5 models submitted MULTICOM_qa" → is a word missing ?

18) Typos:

For parameters, please check:

"num_ensemble_eva" → "num_ensemble_eval"

"num_cycle" → "num_recycle"

"num_ensemble" → "num_ensemble_eval"

"sever_model_dataset" → "server_model_dataset"

"the quality of multimer multimers" (2 times "multimer")

"NBIS-AF2-multimer" and "NIBS-AF2-multimer" → NBIS-AF2-multimer

Github repository :

19) I tried to install MULTICOM3 on a server but didn't manage to have it work properly. The readme is well written and the setup.py file seems to work fine, additional databases are well downloaded but when running the tool, I got some issues. The authors should fix these issues then test a fresh install.

- For monomers, it seems that there is a problem with the name of the fasta file that is not used any more in the following process. I get such an error with a "test_monomer.fasta" file

**

3. Start to generate tertiary structure for monomers using alphafold

Cannot find uniclust30 alignment for test_monomer:

```
(..)/output/N1_monomer_alignments_generation/test_monomer_uniref30_bfd.a3m
```

```
(..)
```

**

I skipped this renaming the fasta file to "test.fasta". But then:

- I got an issue with Mgnify version for monomer, homomer and multimer. It crashes with "mgy_clusters_2022_05.fa" although it is the version set in the "dboption" file. I get this error:

**

Could not find Jackhmmer database (..)/databases//mgnify/mgy_clusters_2018_12.fa

**

20) The default parameters on the Github readme are not exactly the ones of the db_option and .db_option.default files for num_monomer_predictions_per_model and num_multimer_predictions_per_model.

21) Typos in the readme:

"rquired third-party packages" → "required third-party packages"

"Running the monomer/teritary structure prediction pipeline" → "tertiary"

"python setup.py --envidr \$YOUR_PYTHON_ENV --af_dir \$YOUR_ALPHAFOLD_DIR --afdb_dir \$YOUR_ALPHAFOLD_DB_DIR" → should be "--envdir"

"to store the some key parameter values" -> "the" to remove

22) "export PYTHONPATH=MULTICOM3_INSTALL_DIR " → it could be easier for the user to explain

that the path of the MULTICOM3 INSTALL_DIR should be set there because people may simply copy/paste but it wouldn't work

23) I didn't find how to specify AlphaFold parameters such as the -use_precomputed_msas in the Multicom parameters. It should be possible to specify them. Please explain in the readme how to change them.

24) In "Multimer (Homo-multimer and hetero-multimer)", "complex" for monomer data,

"quaternary" for "multimer" data, I think it makes things a bit confusing. Why not

"N4_monomer_alignments_concatenation", "N5_monomer_templates_concatenation" and

"N6_multimer_structure_generation". Also again, using "model" instead of "prediction", I find it a bit confusing.

25) In the logs, I see some paths containing //, it doesn't make the tool crash but it would be better to get simple /.

Response to Review Comments

Reviewer #1

The authors have proposed a new protein prediction tool called MULTICOM3, based on AlphaFold-Multimer. In this work, the authors utilized various methods such as FoldSeek and deepmsa to generate structural templates and MSA (multiple sequence alignment) to enhance AlphaFold-Multimer. This method demonstrated good performance in CASP15.

1. How significant is the role of Pair-MSA in MULTICOM3? Due to the relatively small number of target proteins in CASP15, are there any comparative results with a benchmark dataset available?

Response:

Thank you for the insightful question. To analyze the performance of the all 13 kinds of MSAs listed in Table S4 on protein complexes more thoroughly, we created a new benchmark dataset from the proteins deposited in the PDB after AlphaFold-Multimer was released as follows. We retrieved complex structures from the PDB released between 04/01/2022 and 12/09/2022. The structures were subjected to a series of filtering, considering the criteria such as sequence length (>1536 residues), resolution (>4 Angstrom), and chain number (>8). The complexes were considered hetero-multimers based on a 0.9 sequence identity threshold between chains. Only hetero-multimers were used in this post-CASP15 experiment because all the 13 MSA generation methods were applied to them, while the homo-multimer structure prediction used only 7 kinds of MSAs without the complicated monomer MSA pairing. To remove artificial hetero-multimers due to the experimental artifact, we only retained hetero-multimers whose structure had at least ten inter-chain residue-residue pairs in contact with a minimum distance of < 5 Angstrom between any heavy atoms. To remove the hetero-multimers that are similar to known protein structures that may be used by AlphaFold-Multimer and MULTICOM3, we excluded hetero-multimers whose subunits had more than 0.4 sequence identity with the monomer chains in the PDB prior to 04/01/2022. Additionally, a hetero-multimer was removed if any one of its subunits had a template hit in the monomer template database of MULTICOM3 consisting of monomer structures released by 04/01/2022 by HHSearch at the e-value threshold of 1. Moreover, the subunits of hetero-multimers were clustered using MMseqs2 with a 0.3 sequence identity threshold. The cluster ID assigned to each hetero-multimer was determined by the combination of the cluster IDs of its monomer chains. The hetero-multimer with the best resolution in each cluster was selected to be included into the final benchmark dataset. The dataset has 100 hetero-multimers in total.

For each of the hetero-multimers in the dataset, the MSA_{paired} and the MSA_{unpaired} generated by our in-house default AlphaFold-Multimer were used to generate 25 predictions without using any templates. For a fair comparison, each of the 13 different kinds of paired MSAs of its subunits (MSA_{paired}) generated by MULTICOM3 together with the exactly same unpaired MSAs of the subunits (MSA_{unpaired}) was used by AlphaFold-Multimer to generate 25 predictions, without using any structural template as input. The top 5

predictions for each kind of MSA_{paired} were selected by the AlphaFold-Multimer's confidence score. The results of 13 kinds of MSA and the default AlphaFold-Multimer MSA are shown in supplementary **Figure S2**, where each MSA_{paired} is named by its interaction source and sequence database.

In supplementary **Figure S2A**, the average TM-score of the top-1 predictions on the 100 hetero-multimers for the MSA_{paired} generated by the default AlphaFold-Multimer (denote as `default_multimer`) is 0.799, which is higher than the average score of the other 13 kinds of MSA_{paired} ranging from 0.7697 to 0.788. However, according to the one-sided Wilcoxon signed rank test, there is no significant difference between `default_multimer` and 7 kinds of MSA_{paired} (`spec_iter_uniprot_sto`, `str_iter_uniref_sto`, `str_iter_uniprot_sto`, `unidist_uniprot_sto`, `pdb_iter_uniref_sto`, `pdb_iter_uniref_a3m`, and `unidist_uniref_a3m`). The average TM-score of the best of top 5 predictions of `default_multimer` is 0.8206, higher than the average score of the 13 MSAs ranging from 0.7954 to 0.8153, but there is no significant difference between `default_multimer` and 5 kinds of MSA_{paired} (`str_iter_uniref_sto`, `str_iter_uniprot_sto`, `spec_iter_uniprot_sto`, `pdb_iter_uniref_a3m`, and `str_iter_uniref_a3m`) (supplementary **Figure S2B**).

To further investigate the effectiveness of combining the 13 kinds of MSA_{paired} , the `default_multimer` was employed to generate 325 predictions for each hetero-multimer, which were used to compare with the 325 predictions in the combined prediction pool of the 13 kinds of MSA_{paired} (denote as *combine*). Notably, the average TM-score of the *combine* method for top 1 (or best of top 5) predictions selected by the AlphaFold-Multimer's confidence score is 0.8045 (or 0.8317), higher than the 0.7997 (or 0.8212) of `default_multimer` (supplementary **Figure S2C**), even though the difference is not statistically significant. The results show that even though each of the 13 kinds of MSAs does not perform better than the default MSA, the predictions generated from them as whole have better quality than the default MSA, demonstrating their complementarity across different targets. Interestingly, only increasing the number of predictions from 25 to 325 for the `default_multimer` resulted in a much smaller improvement (e.g., 0.0004 TM-score difference for top 1 predictions and 0.0006 TM-score difference for top five predictions). The results show that sampling predictions with diverse MSAs can improve the quality of the assembly structure prediction more substantially than only increasing the number of predictions generated.

We have added the new results and analysis into Section "Sampling predictions with diverse multiple sequence alignments and templates improves assembly structure prediction".

2. The authors mentioned using various Pair-MSAs, but it is not clear how they were generated. The open-source code provided on GitHub was unable to successfully run and generate Pair-MSA for the test cases provided by the authors.

Response:

Thank you for noticing the problem. We have fixed the problem on GitHub and tested it thoroughly (see new version v2.0.0 of MULTICOM3 at GitHub). Please follow the instructions in the readme to run the test cases with the latest version of MULTICOM3. More detailed description about how paired MSAs were generated was added into the manuscript.

3. Regarding the results obtained from FoldSeek, are all the results used to enhance MSA, or are there specific strategies to select the good results for enhancing MSA? If there are selection strategies, please provide details.

Response:

There are specific strategies to select the structural alignments for enhancing MSA. The version of Foldseek we used is the first released version (1-3c64211). There are two kinds of structural alignment that can be obtained by setting the parameter (--alignment-type) to 1 or 2, where 1 is the global alignment calculated by TAlign, and 2 is the local alignment by 3Di+AA Gotoh-Smith-Waterman.

For the Foldseek Structure Alignment-based Multimer structure Generation (FSAMG), The structural alignments of local structural hits with e-value ≤ 0.001 and global structural hits with TM-score ≥ 0.3 are converted into the sequence alignments between the target and the hits.

For hetero-multimers (supplementary **Figure S5A**), the $MSA_{unpaired}$ is initialized as the MSA generated for each subunit by the tertiary structure prediction system, while the MSA_{paired} is set empty. Two sequence alignments generated from structural alignments for two subunits/chains of the hetero-multimer are paired if they come from the same PDB protein complex (i.e., sharing the same PDB code but from different chains) or from the two non-overlapping domains of a hit in the AlphaFoldDB. Only one paired alignment is kept for each PDB code to avoid the redundancy in the paired alignments. The sequence alignments are added into the $MSA_{unpaired}$ for each subunit, while the paired sequence alignments are included into the MSA_{paired} .

For homo-multimers (supplementary **Figure S5B**), two different approaches are applied to generate the MSA_{paired} . The first approach initializes the MSA_{paired} with the horizontal concatenation of the multiple copies of the MSA of monomer generated by the tertiary structure prediction system. Subsequently, the sequence alignments of individual subunits/chains that share the same PDB code but from different chains or from the non-overlapping domains of a hit in the AlphaFoldDB are paired. The top 50 paired alignments ranked by e-value and TM-score are added into the MSA_{paired} . The second approach initializes the MSA_{paired} by the pairings of sequence alignments generated from the structural alignments that share the same PDB code or ID in the AlphaFoldDB. Then the horizontal concatenations of the multiple copies of the MSA of monomer generated by the tertiary structure prediction system are added to the MSA_{paired} .

For the Foldseek Structure Alignment-based Multimer structure refinement, the structure alignments with evalue ≤ 0.001 or TM-score ≥ 0.3 between each subunit and structural hits are converted into sequence alignments. The sequence alignments of the subunits generated from the Foldseek search are concatenated if they are from the same PDB complex structure or the non-overlapped regions of the same single-chain AlphaFoldDB prediction to construct the MSA for the multimer. The top structural hits of the subunits are used as the monomer templates for each subunit of the multimer. The concatenated MSAs are ranked by e-value and TM-score. Only the top 50 concatenated MSAs are added to the original MSA_{paired} to generate a *deeper* MSA.

We have added the details above to the **Materials and Methods** section.

4. How do you convert a structural alignment between two proteins into a sequence alignment?

Response:

Thank you for raising the question. The version of Foldseek we used is the first released version (1-3c64211). There are two kinds of structural alignment that can be obtained by setting the parameter (--alignment-type) to 1 or 2, where 1 is the global alignment calculated by TMAAlign, and 2 is the local alignment by 3Di+AA Gotoh-Smith-Waterman. The output of Foldseek will return the start (end) position of the alignment in the query (qstart/qend), and the start (end) position of the alignment in the template (tstart, tend), and the pairwise alignments between the query and the template (qaln and taln). For the global alignments, the structural alignment can be directly used as sequence alignment between the input sequence from the predicted structure and the sequence from the template as it is calculated by the TMAAlign. For the local alignments, we convert the local structural alignment into sequence alignment by adding gaps for the regions where the template didn't cover.

5. For homomeric complexes, if the results obtained from FoldSeek for subunits do not meet the condition "Paired by PDB code or non-overlapping regions of AlphaFoldDB hit" as shown in Figure S3, does the FSAMG method only provide structural templates for individual subunits without the enhanced MSA component? If that is the case, can the performance of multimers still be improved in such situations?

Response:

Thank you for your insightful comment. First, we would like to clarify that there are two different approaches are applied to generate the MSA_{paired} for homo multimers. The first approach initializes the MSA_{paired} with the horizontal concatenation of the multiple copies of the MSA of monomer generated by the tertiary structure prediction system. Subsequently, the sequence alignments of individual subunits/chains that share the same PDB code but from different chains or from the non-overlapping domains of a hit in the AlphaFoldDB are paired. The top 50 paired alignments ranked by e-value and TM-score are added into the MSA_{paired} . The second approach initializes the MSA_{paired} by the the pairings of sequence alignments generated from the structural alignments that share the same PDB code or ID in the AlphaFoldDB. Then the horizontal concatenations of the multiple copies of the MSA of monomer generated by the tertiary structure prediction system are added to the MSA_{paired} .

For each subunit in the hetero-multimer and homo-multimer, the similar structural hits from the pdb_complex template database and the AlphaFoldDB are ranked by evalue and TM-score. The ranked structural hits are treated as monomer templates for each subunit. The structure-alignment generated MSAs and the monomer templates for each subunit are used as input for the customized AlphaFold-Multimer to generate 10 predictions.

As for the first strategy, you are correct that this strategy only provides structural templates for individual subunits without the enhanced MSA component when no paired alignments can be found by Foldseek.

However, the structure prediction generated by this strategy can still be improved over the standard AlphaFold-Multimer, especially for T1173o. The reason is that FSAMG found 4 significant monomer templates including 4UW7A, 4UW4B, 4UW7C from a homo-trimer 4UW7, and 5AQ5B from another homo-trimer 5AQ5 for T1173o that were used as input for AlphaFold-Multimer to generate predictions. The proportion of high-accuracy predictions (TM-score > 0.95) generated by FSAMG is 60% (**Figure 6**), while NBIS-AF2-multimer and our other AlphaFold-Multimer variants did not generate any prediction of such high accuracy. This example demonstrates that the performance of multimeters can still be improved in such situations (e.g., without enhanced MSA component).

The second strategy has ensured that the enhanced MSA component and the structural templates are supplied to AlphaFold-Multimer to generate predictions for all the 15 homomeric complexes out of 31 complexes.

We have added the necessary technical details of generating the alignments into the manuscript.

Reviewer #2

Jian Liu et al present a tool for quaternary structure prediction called MULTICOM, which is an improved version of the famous AlphaFold-Multimer prediction. In particular, they enhanced the MSA generation to provide to AlphaFold-Multimer, used a new complementary metric for ranking predictions and added a new refinement method based on Foldseek. The improvements in the quality of the predictions don't look spectacular with about 8% higher than AlphaFold on the TM-score, which means in general, the folding and interface should be conserved as shown in the CAPRI evaluation of the CASP15 round. However, the authors give here a fair presentation of the results, showing when their tool really improve the results and when it doesn't, with not exaggerated improvement. In addition, with AlphaFold-Multimer, the predictions are at a high level because the tool with default parameters already provide very nice predictions for many targets. Therefore, improving its predictions is challenging and any enhancement is always an added-value for the scientific community, knowing moreover that MULTICOM very rarely degrades the predictions while, in the majority of cases, don't change or improve them. The authors also provide the tool on a github repository for the last version of AlphaFold-Multimer with details for installing and running it. I would recommend the publication but suggest a few points to be addressed to improve the manuscript and some bugs have to be fixed on the Github repository.

1) The beginning of the introduction (second paragraph) could be improved a bit about classical historical docking methods. "the accuracy of the protein docking is generally low" could be put into perspective with cases where these methods work well and cases where they fail. The references 1 and 2 that are given are not the most appropriate because they refer to protein-ligand methods while the article is only based on protein complex prediction. These references should be replaced by more appropriate and more recent ones.

Response:

Thank you for the great suggestion. We have rewritten the paragraph in the **Introduction** to better describe the methods in protein complex structure prediction as follows.

“Traditionally, the prediction of protein complex structures employs template-based modeling or ab-initio methods such as protein-protein docking. In template-based modeling, complex templates with known structures are initially identified for a target protein complex and subsequently utilized to construct its structural prediction. While this approach proves effective if similar homologous templates can be found, it does not work for most targets because good templates are usually not available or cannot be identified. Different from the template-based modeling, ab-initio methods often strive to generate complex structures from scratch for a target through simulation techniques (e.g., simulated annealing or Markov Chain Monte Carlo simulation) by optimizing score functions (e.g., energy or statistical potential functions^{1,2} or machine learning scoring^{3,4}) measuring the quality of complex structural predictions, given the tertiary structures of the individual subunits of the target as input. However, these methods are computationally demanding due to the extensive conformation space they must explore. The accuracy of the most studied ab initio method - protein docking is generally low^{5,6}, particularly when confronted with suboptimal input tertiary structures.”

Moreover, References 1 and 2 were replaced and some new references were added as follows.

1. Trott O, Olson AJ. AutoDock Vina: improving the speed and accuracy of docking with a new scoring function, efficient optimization, and multithreading. *Journal of computational chemistry* 2010;31:455-461.
2. Friesner RA, Banks JL, Murphy RB, et al. Glide: a new approach for rapid, accurate docking and scoring. 1. Method and assessment of docking accuracy. *Journal of medicinal chemistry* 2004;47:1739-1749.
3. McNutt AT, Francoeur P, Aggarwal R, et al. GNINA 1.0: molecular docking with deep learning. *Journal of cheminformatics* 2021;13:1-20.
4. Méndez-Lucio O, Ahmad M, del Rio-Chanona EA, et al. A geometric deep learning approach to predict binding conformations of bioactive molecules. *Nature Machine Intelligence* 2021;3:1033-1039.
5. Lensink MF, Brysbaert G, Nadzirin N, et al. Blind prediction of homo-and hetero-protein complexes: The CASP13-CAPRI experiment. *Proteins: Structure, Function, and Bioinformatics* 2019;87:1200-1221.
6. Lensink MF, Brysbaert G, Mauri T, et al. Prediction of protein assemblies, the next frontier: The CASP14-CAPRI experiment. *Proteins: Structure, Function, and Bioinformatics* 2021;89:1800-1823.

2) “Results and conclusion” section should be renamed “Results and discussion” and although the sections are well explained, they could sometimes be put a bit more into perspective about for instance:

- what are the cases that still fail for AlphaFold with suggestions to improve these cases
- the effect of the extended sampling because with MULTICOM, the number of generated predictions is higher than the 25 predictions of AlphaFold-Multimer by default

Response:

Thanks for the great suggestions. We have renamed the “Results and conclusion” section as “Results and discussion”.

Regarding the first question, it is worth noting that AlphaFold-Multimer was rather effective in generating structural predictions for small complexes with diverse sampling strategies in CASP15. For the small complexes (e.g., less than 6 chains), extensive sampling approaches (such as generating over 1000 predictions) through AlphaFold-Multimer employed by some groups such as the Wallner group yielded some high-quality structural predictions. The main challenge for these approaches lies in selecting the structural prediction with highest quality, which can be tackled by developing more effective methods for assessing the quality of multimer structures.

However, the prediction complexity intensifies when dealing with large higher-order complexes due to two main factors. Firstly, predicting the multimer structure for higher-order complexes demands substantial computational resources (e.g., H1111, H1114, T1115o, H1137) that may not be available. In this case, dividing a large multimer into subcomplexes to generate predictions for them to be combined into full-length predictions for the multimer is a viable option. However, a large multimer usually has too many sub-complexes to generate predictions for in a limited amount of time. Identifying critical sub-complexes that can link sub-complexes together to form the structure of the entire multimer is critical and sometime very challenging. For instance, constructing full-length structures for H1111 (A9B9C9) and H1114 (A4B8C8) hinges on generating the structures of subcomplex C9 of H1111 and subcomplex A4 of H1114 because they form the backbone of linking all the chains in the multimers together. However, it is not always obvious to prioritize them for structure generation. Secondly, higher-order complexes may have multiple protein-protein interaction interfaces between their units, which present a challenge for AlphaFold-Multimer to predict and for combining them to generate full-length multimer structures due to a large number of combinatorial choices. For example, there are more than four possible interaction interfaces between two identical subunits of T1176o (A8) predicted by AlphaFold-Multimer. No full-length prediction for this target submitted from the CASP15 predictors has TM-score higher than 0.5. Addressing these challenges may require the development of efficient and effective methods for the combinatorial problems of selecting subcomplexes to generate predictions and combining different predictions with different interfaces to generate full-length structures of multimers.

We have added the discussion in section **Prediction of the structures of very large assemblies**.

For the second question, to analyze the performance of the all 13 kinds of MSAs in **Table S4** on protein complexes more thoroughly, after CASP15 was concluded, we created a new benchmark dataset from the proteins deposited in the PDB after AlphaFold-Multimer was released as follows. We retrieved complex structures from the PDB released between 04/01/2022 and 12/09/2022. The structures were subjected to a series of filtering, considering the criteria such as sequence length (>1536 residues), resolution (>4 Angstrom), and chain number (>8). The complexes were considered hetero-multimers based on a 0.9 sequence identity threshold between chains. Only hetero-multimers were used in this post-CASP15 experiment because all the 13 MSA generation methods were applied to them, while the homo-multimer structure prediction used only 7 kinds of MSAs without the complicated monomer MSA pairing. To remove artificial hetero-multimers due to the experimental artifact, we only retained hetero-multimers whose structure had at least ten inter-chain residue-residue pairs in contact with a minimum distance of < 5 Angstrom between any heavy atoms. To remove the hetero-multimers that are similar to known protein

structures that may be used by AlphaFold-Multimer and MULTICOM3, we excluded hetero-multimers whose subunits had more than 0.4 sequence identity with the monomer chains in the PDB prior to 04/01/2022. Additionally, a hetero-multimer was removed if any one of its subunits had a template hit in the monomer template database of MULTICOM3 consisting of monomer structures released by 04/01/2022 by HHSearch at the e-value threshold of 1. Moreover, the subunits of hetero-multimers were clustered using MMseqs2 with a 0.3 sequence identity threshold. The cluster ID assigned to each hetero-multimer was determined by the combination of the cluster IDs of its monomer chains. The hetero-multimer with the best resolution in each cluster was selected to be included into the final benchmark dataset. The dataset has 100 hetero-multimers in total.

For each of the hetero-multimers in the dataset, the MSA_{paired} and the MSA_{unpaired} generated by our in-house default AlphaFold-Multimer were also used to generate 25 predictions without using any templates. For a fair comparison, each of the 13 different kinds of paired MSAs of its subunits (MSA_{paired}) generated by MULTICOM3 together with the exactly same unpaired MSAs of the subunits (MSA_{unpaired}) was used by AlphaFold-Multimer to generate 25 predictions, without using any structural template as input. The top 5 predictions for each kind of MSA_{paired} were selected by the AlphaFold-Multimer's confidence score. The results of 13 kinds of MSA and the default AlphaFold-Multimer MSA are shown in supplementary **Figure S2**, where each MSA_{paired} is named by its interaction source and sequence database.

In supplementary **Figure S2A**, the average TM-score of the top-1 predictions on the 100 hetero-multimers for the MSA_{paired} generated by the default AlphaFold-Multimer (denote as *default_multimer*) is 0.799, which is higher than the average score of the other 13 kinds of MSA_{paired} ranging from 0.7697 to 0.788. However, according to the one-sided Wilcoxon signed rank test, there is no significant difference between *default_multimer* and 7 kinds of MSA_{paired} (*spec_iter_uniprot_sto*, *str_iter_uniref_sto*, *str_iter_uniprot_sto*, *unidist_uniprot_sto*, *pdb_iter_uniref_sto*, *pdb_iter_uniref_a3m*, and *unidist_uniref_a3m*). The average TM-score of the best of top 5 predictions of *default_multimer* is 0.8206, higher than the average score of the 13 MSAs ranging from 0.7954 to 0.8153, but there is no significant difference between *default_multimer* and 5 kinds of MSA_{paired} (*str_iter_uniref_sto*, *str_iter_uniprot_sto*, *spec_iter_uniprot_sto*, *pdb_iter_uniref_a3m*, and *str_iter_uniref_a3m*) (supplementary **Figure S2B**).

To further investigate the effectiveness of combining the 13 kinds of MSA_{paired} , the *default_multimer* was employed to generate 325 predictions for each hetero-multimer, which were used to compare with the 325 predictions in the combined prediction pool of the 13 kinds of MSA_{paired} (denote as *combine*). Notably, the average TM-score of the *combine* method for top 1 (or best of top 5) predictions selected by the AlphaFold-Multimer's confidence score is 0.8045 (or 0.8317), higher than the 0.7997 (or 0.8212) of *default_multimer* (supplementary **Figure S2C**), even though the difference is not statistically significant. The results show that even though each of the 13 kinds of MSAs does not perform better than the default MSA, the predictions generated from them as whole have better quality than the default MSA, demonstrating their complementarity across different targets. Interestingly, only increasing the number of predictions from 25 to 325 for the *default_multimer* resulted in a much smaller improvement (e.g., 0.0004 TM-score difference for top 1 predictions and 0.0006 TM-score difference for top five predictions).

The results show that sampling predictions with diverse MSAs can improve the quality of the assembly structure prediction more substantially than only increasing the number of predictions generated.

We have added the discussion in Section “**Sampling predictions with diverse multiple sequence alignments and templates improves assembly structure prediction**”.

3) Do the authors know which combination of MSA work the best ? If they have access to such an evaluation, they could precise it in the text.

Response:

Thank you for the great suggestion. We conducted a new experiment to answer this question. To analyze the performance of the all 13 kinds of MSAs in **Table S4** on protein complexes more thoroughly, after CASP15 was concluded, we created a new benchmark dataset from the proteins deposited in the PDB after AlphaFold-Multimer was released as follows. We retrieved complex structures from the PDB released between 04/01/2022 and 12/09/2022. The structures were subjected to a series of filtering, considering the criteria such as sequence length (>1536 residues), resolution (>4 Angstrom), and chain number (>8). The complexes were considered hetero-multimers based on a 0.9 sequence identity threshold between chains. Only hetero-multimers were used in this post-CASP15 experiment because all the 13 MSA generation methods were applied to them, while the homo-multimer structure prediction used only 7 kinds of MSAs without the complicated monomer MSA pairing. To remove artificial hetero-multimers due to the experimental artifact, we only retained hetero-multimers whose structure had at least ten inter-chain residue-residue pairs in contact with a minimum distance of < 5 Angstrom between any heavy atoms. To remove the hetero-multimers that are similar to known protein structures that may be used by AlphaFold-Multimer and MULTICOM3, we excluded hetero-multimers whose subunits had more than 0.4 sequence identity with the monomer chains in the PDB prior to 04/01/2022. Additionally, a hetero-multimer was removed if any one of its subunits had a template hit in the monomer template database of MULTICOM3 consisting of monomer structures released by 04/01/2022 by HHSearch at the e-value threshold of 1. Moreover, the subunits of hetero-multimers were clustered using MMseqs2 with a 0.3 sequence identity threshold. The cluster ID assigned to each hetero-multimer was determined by the combination of the cluster IDs of its monomer chains. The hetero-multimer with the best resolution in each cluster was selected to be included into the final benchmark dataset. The dataset has 100 hetero-multimers in total.

For each of the hetero-multimers in the dataset, the MSA_{paired} and the MSA_{unpaired} generated by our in-house default AlphaFold-Multimer were also used to generate 25 predictions without using any templates. For a fair comparison, each of the 13 different kinds of paired MSAs of its subunits (MSA_{paired}) generated by MULTICOM3 together with the exactly same unpaired MSAs of the subunits (MSA_{unpaired}) was used by AlphaFold-Multimer to generate 25 predictions, without using any structural template as input. The top 5 predictions for each kind of MSA_{paired} were selected by the AlphaFold-Multimer’s confidence score. The results of 13 kinds of MSA and the default AlphaFold-Multimer MSA are shown in supplementary **Figure S2**, where each MSA_{paired} is named by its interaction source and sequence database.

In supplementary **Figure S2A**, the average TM-score of the top-1 predictions on the 100 hetero-multimers for the MSA_{paired} generated by the default AlphaFold-Multimer (denote as `default_multimer`) is 0.799, which is higher than the average score of the other 13 kinds of MSA_{paired} ranging from 0.7697 to 0.788. However, according to the one-sided Wilcoxon signed rank test, there is no significant difference between `default_multimer` and 7 kinds of MSA_{paired} (`spec_iter_uniprot_sto`, `str_iter_uniref_sto`, `str_iter_uniprot_sto`, `unidist_uniprot_sto`, `pdb_iter_uniref_sto`, `pdb_iter_uniref_a3m`, and `unidist_uniref_a3m`). The average TM-score of the best of top 5 predictions of `default_multimer` is 0.8206, higher than the average score of the 13 MSAs ranging from 0.7954 to 0.8153, but there is no significant difference between `default_multimer` and 5 kinds of MSA_{paired} (`str_iter_uniref_sto`, `str_iter_uniprot_sto`, `spec_iter_uniprot_sto`, `pdb_iter_uniref_a3m`, and `str_iter_uniref_a3m`) (supplementary **Figure S2B**).

To further investigate the effectiveness of combining the 13 kinds of MSA_{paired} , the `default_multimer` was employed to generate 325 predictions for each hetero-multimer, which were used to compare with the 325 predictions in the combined prediction pool of the 13 kinds of MSA_{paired} (denote as *combine*). Notably, the average TM-score of the *combine* method for top 1 (or best of top 5) predictions selected by the AlphaFold-Multimer's confidence score is 0.8045 (or 0.8317), higher than the 0.7997 (or 0.8212) of `default_multimer` (supplementary **Figure S2C**), even though the difference is not statistically significant. The results show that even though each of the 13 kinds of MSAs does not perform better than the default MSA, the predictions generated from them as whole have better quality than the default MSA, demonstrating their complementarity across different targets. Interestingly, only increasing the number of predictions from 25 to 325 for the `default_multimer` resulted in a much smaller improvement (e.g., 0.0004 TM-score difference for top 1 predictions and 0.0006 TM-score difference for top five predictions). The results show that sampling predictions with diverse MSAs can improve the quality of the assembly structure prediction more substantially than only increasing the number of predictions generated.

We have added the discussion in Section **“Sampling predictions with diverse multiple sequence alignments and templates improves assembly structure prediction”**.

4) In “The comparison of the multimer model quality assessment methods” section, the AlphaFold-Multimer confidence score is used. Because this score is based for 80% on the ipTM, I am curious to see what the loss and correlation would be comparing to the DockQ that is more focused on the interface, rather than the TM-score.

Response:

Thank you for the great suggestion. We have added the evaluation in terms of DockQ score on the `server_prediction_dataset` and `human_prediction_dataset` into **Table 2**.

On the `server_prediction_dataset`, Confidence has the lowest average ranking loss of DockQ loss (0.1003), while CoPSS has the highest average correlation (0.4073), indicating that they have some complementarity. On the `human_prediction_dataset`, Confidence has the lowest average ranking loss of

DockQ score (0.0979), while CoPSS has the highest correlation (0.459). Based on the results on the two datasets, Confidence and PSS are complementary for estimating the accuracy of multimer predictions, while Confidence performs better in estimating the interface accuracy of the multimer predictions in terms of the loss of DockQ score. Combining them may be useful to improve the quality assessment of multimer predictions. However, how to combine them to achieve consistently better results still needs more investigation.

We have added the new results and discussion above into Section **“The comparison of the multimer structure quality assessment methods”**.

5) The term “model” is a bit vague, especially now that we talk about neural network models in structural bioinformatics. I would suggest to use the term “prediction” to avoid any confusion with any other type of model.

Response:

Thanks for pointing out the confusion. We have used “prediction” or “structure” to replace “model” to avoid the confusion in main text and the supplementary materials.

6) The version of AlphaFold-Multimer that was used for the article should be precised in the “Materials and Methods” section

Response:

Thank you for the comment. We have specified the version of AlphaFold-Multimer and AlphaFold2 in the **“Materials and Methods”** section as v2.2.0.

7) Figure 1 for T1174o: are the authors sure about the TM-scores ? Because the structures look similar but the TM-scores are very different. If this is correct, could they explain why such a difference ?

Response:

Thank you for raising this question. The results of TM-scores are downloaded from the CASP15 website (https://predictioncenter.org/casp15/multimer_results.cgi?target=T1174o). We double checked the TM-scores between the native structure and the predictions of MULTICOM_qa or Manifold-E using USAlign. The results are consistent with the official CASP results. However, the TM-score between the two predictions is 0.99, indicating they are very similar. We believe the reason that MULTICOM_qa predicted structure has a much lower TM-score than Manifold-E predicted structure is that USAlign sometimes generates suboptimal chain assignments (mapping) when comparing a prediction to the native structure occasionally as stated in their paper (<https://www.nature.com/articles/s41592-022-01585-1>). In this case, the chain mapping between the MULTICOM_qa prediction and the native structure generated by USAlign is worse than that between the Manifold-E prediction and the native structure, resulting in a lower TM-score for the former.

8) Part "Overall performance of MULTICOM_qa compared with the standard AlphaFold-Multimer": that would be nice to show the structures H1114, T1176o, T1115o, T1160o and T1161o like in Figure 1 in Supplementary materials for instance

Response:

Thank you for the great suggestion. We have added new **Figure S1** in the supplementary material to show the best structural prediction submitted from MULTICOM_qa and the native structures for H1114, T1176o, T1115o, T1160o and T1161o.

9) For FASMG, why 26 complexes and not more ?

Response:

Thank you for bringing up the question. For the total number of 36 complexes with experimental structures available, we only ran FASMG on the 26 complexes due to the server prediction deadline. Our computational resources were limited, preventing us from completing FASMG on each CASP15 complex within the 3-day time limit.

10) In "Materials and Methods" section: String interaction scores are between 0 and 1, then could the authors explain how the 500 threshold was obtained ?

Response:

Thanks for pointing out the problem. The version of the String database we used in the MULTICOM3 is v11.0, not the latest version v11.5. As you stated, the String interactions scores are between 0 and 1. However, in their files they have multiplied the interaction scores by 1000 to make them integers (see <https://version-11-0b.string-db.org/cgi/help?sessionId=bOyxEkNpmvFV> and search for "How to extract high confidence (>0.7) interactions from information on "combined score" in "protein.links.txt.gz"). Therefore, we have used 500 here. We have specified the version of the String database and the threshold should be 0.5 in the main text.

11) The differences between all the MULTICOM_xxx versions are not always clear. I suggest a table that explains the differences for each version in Supplementary Materials

Response:

Thank you for the suggestion. We have added the following table (**Table S5**) to Supplementary Materials to explain the differences for each version of MULTICOM server predictors and human predictors.

12) What are the differences between the Github version and the one used in the paper ? I see that the last AlphaFold version (2.3.2) that the user installs can be used, which is very good, but are there any other differences because the article says that a customized version of AlphaFold was used ?

Response:

Thanks for pointing out the confusion. At the GitHub repository, AlphaFold from DeepMind has been upgraded from v2.2.0 used in this work to v2.3.2 so that users can take advantage of the latest improvement of AlphaFold2. However, our MULTICOM3 add-on package at the GitHub is the same as the one used in this work.

13) What is the best configuration (memory/CPU) to run the tool ? That could be useful to add it in the Materials and Methods section so that the user knows for its installation.

Response:

Thanks for the great suggestion. During CASP15, the MULTICOM protein assembly structure prediction system was mainly executed on three computer servers (server 1: 192 AMD EPYC 7552 48-Core CPU Processor, 377 GB RAM, an NVIDIA A100 PCIe 80GB GPU; server 2: 192 AMD EPYC 7552 48-Core CPU Processor, 503 GB RAM, an NVIDIA A100 PCIe 40GB GPU; and server 3: 192 AMD EPYC 7552 48-Core CPU Processor, 1 TB RAM, an NVIDIA A100 PCIe 40GB GPU) respectively to generate the predictions. Users can use the similar hardware to run the tool.

We have added the configuration (memory/CPU) to the sub-section “**Implementation of the CASP15 assembly structure predictors**” under **Materials and Methods** section.

Minor points:

14) Sometimes the term “AlphaFold-Multimer” is used and sometimes “AlphaFold2-Multimer” is used, one of both should be chosen and used everywhere. In addition, please check the spelling (e.g.: AlphaFold-Multimer)

Response:

Thanks for pointing out the problem. We confirm “AlphaFold-Multimer” is the only term referring to AlphaFold-Multimer in the main text. We have made the terms consistent in the main manuscript.

15) CASP is “Critical Assessment of protein Structure Prediction”

Response:

Thank you for the comment. We defined the term CASP as “Critical Assessment of Techniques for Protein Structure Prediction” based on its official website (<https://predictioncenter.org/casp15/index.cgi>).

16) In “Overall performance of MULTICOM_qa compared with the standard AlphaFold-Multimer” section: “(H1111, T1187o, T1173o, T1115o, H1135, T1181o, T1123o, T1179o, H1137)” is repeated three times. Please avoid repetitions in this part.

Response:

Thanks for pointing out the repetitions. We have adjusted the contents to avoid the repetitions of the nine targets.

17) "For instance, for T1174o and T1181o, there were two alternative conformations in the top 5 models submitted MULTICOM_qa" → is a word missing ?

Response:

Thanks for pointing out the problem. We have fixed the problem.

18) Typos:

For parameters, please check:

"num_ensemble_eva" → "num_ensemble_eval"

"num_cycle" → "num_recycle"

"num_ensemble" → "num_ensemble_eval"

"sever_model_dataset" → "server_model_dataset"

"the quality of multimer multimers" (2 times "multimer")

"NBIS-AF2-mutlimer" and "NIBS-AF2-multimer" → NBIS-AF2-multimer

Response:

Thanks for pointing out the typos. We have fixed the problem.

Github repository :

19) I tried to install MULTICOM3 on a server but didn't manage to have it work properly. The readme is well written and the setup.py file seems to work fine, additional databases are well downloaded but when running the tool, I got some issues. The authors should fix these issues then test a fresh install.

- For monomers, it seems that there is a problem with the name of the fasta file that is not used any more in the following process. I get such an error with a "test_monomer.fasta" file

**

3. Start to generate tertiary structure for monomers using alphafold

Cannot find uniclust30 alignment for test_monomer:

(..)/output/N1_monomer_alignments_generation/test_monomer_uniref30_bfd.a3m

(..)

**

Response:

Thanks for pointing out the problem. We have fixed the problem in the system. For simplicity, we recommend making the name of the sequence file in FASTA format the same as the sequence name. We have added the instructions to the readme on Github.

I skipped this renaming the fasta file to “test.fasta”. But then:

- I got an issue with Mgnify version for monomer, homomer and multimer. It crashes with “mgy_clusters_2022_05.fa” although it is the version set in the “dboption” file. I get this error:

**

Could not find Jackhammer database (..)/databases//mgnify/mgy_clusters_2018_12.fa

**

Response:

Thanks for pointing out the problem. We have fixed the problem in the latest version of MULTICOM3 (v2.0.0).

20) The default parameters on the Github readme are not exactly the ones of the db_option and .db_option.default files for num_monomer_predictions_per_model and num_multimer_predictions_per_model.

Response:

Thanks for pointing out the discrepancy. We have adjusted the parameters in the “.db_option.default” file.

21) Typos in the readme:

“rquired third-party packages” → “required third-party packages”

“Running the monomer/teritary structure prediction pipeline” → “tertiary”

“python setup.py --envidr \$YOUR_PYTHON_ENV --af_dir \$YOUR_ALPHAFOLD_DIR --afdb_dir \$YOUR_ALPHAFOLD_DB_DIR” → should be “--envdir”

“to store the some key parameter values” -> “the” to remove

Response:

Thanks for pointing out the typos. We have fixed the problem.

22) “export PYTHONPATH=MULTICOM3_INSTALL_DIR “ → it could be easier for the user to explain that the path of the MULTICOM3 INSTALL_DIR should be set there because people may simply copy/paste but it wouldn’t work

Response:

Thanks for the great suggestion. We have added more instructions for the user to explain the path of the MULTICOM3_INSTALL_DIR should be set here.

23) I didn't find how to specify AlphaFold parameters such as the `-use_precomputed_msas` in the Multicom parameters. It should be possible to specify them. Please explain in the readme how to change them.

Response:

Thanks for pointing out the confusion. We have set the parameter "use_precomputed_msas" to "True" in the add-on package since the input MSAs are generated in "N1_monomer_alignments_generation" before generating the predictions using AlphaFold2 (N3_monomer_structure_generation) for monomer/subunits of multimer, and "N4_monomer_alignments_concatenation" before generating predictions using AlphaFold-Multimer (N6_multimer_structure_generation).

24) In "Multimer (Homo-multimer and hetero-multimer)", "complex" for monomer data, "quaternary" for "multimer" data, I think it makes things a bit confusing. Why not "N4_monomer_alignments_concatenation", "N5_monomer_templates_concatenation" and "N6_multimer_structure_generation". Also again, using "model" instead of "prediction", I find it a bit confusing.

Response:

Thanks for pointing out the confusion. We have used "prediction" or "structure" to replace "model" to avoid the confusion in the readme file on GitHub.

25) In the logs, I see some paths containing `//`, it doesn't make the tool crash but it would be better to get simple `/`.

Response:

Thanks for pointing out the problem. We have fixed the problem in the latest version of MULTICOM3 (v2.0.0).

Reviewer #1 (Remarks to the Author):

Agree to publish

Reviewer #2 (Remarks to the Author):

The authors answered to all my comments. I reply here to their answers, "OK" meaning I don't have anything else to reply.

1) The 4 first references are not really appropriate in the context of protein-protein docking and references connected to protein-protein docking should replace them.

2) OK

3) OK

4) I would slightly moderate "For DockQ score that specifically considers the quality of the interface", adding a "more" before "specifically", considering the LRMS part of the DockQ computation.

5) OK

6) OK

7) OK

8) OK

9) OK

10) OK

11) OK

12) OK

13) Thanks for details of the configurations. My question was however rather to know if the authors recommend a minimum amount in RAM in particular.

14) OK

15) OK

16) OK

17) OK

18) OK

19) The install and tests worked fine for monomers. I got all the files till N5, without errors in the logs.

But for homomers and heteromers, it crashes at some point. For homomers, it crashes at step 6:

Here is my input fasta file homomer.fasta:

```
>test
```

```
PIAQIHILEGRSDEQKETLIREVSEAIRSLDAPLTSVRVIITEMAKGHFGIGGELASK
```

```
>test
```

```
PIAQIHILEGRSDEQKETLIREVSEAIRSLDAPLTSVRVIITEMAKGHFGIGGELASK
```

and the logs where it crashes:

```
python run_alphaFold_pre.py --fasta_path /multicom_runs/input/homomer.fasta --  
bfd_uniref_a3ms  
/multicom_runs/output/homomer/N1_monomer_alignments_generation/A/A_uniref30_bfd.a3m,/m  
ulticom_runs/output/homomer/N1_monomer_alignments_generation/B/B_uniref30_bfd.a3m --  
mgnify_stos  
/multicom_runs/output/homomer/N1_monomer_alignments_generation/A/A_mgnify.sto,/multicom  
_runs/output/homomer/N1_monomer_alignments_generation/B/B_mgnify.sto --uniref90_stos  
/multicom_runs/output/homomer/N1_monomer_alignments_generation/A/A_uniref90.sto,/multico  
m_runs/output/homomer/N1_monomer_alignments_generation/B/B_uniref90.sto --uniprot_stos  
/multicom_runs/output/homomer/N1_monomer_alignments_generation/A/A_uniprot.sto,/multico  
m_runs/output/homomer/N1_monomer_alignments_generation/B/B_uniprot.sto --env_dir  
/miniconda3/envs/multicom-2.0.2/bin/ --database_dir /shared/databases/2023-07-31 --  
num_multimer_predictions_per_model 5 --multimer_num_ensemble 1 --multimer_num_recycle 3  
--output_dir
```

```
/multicom_runs/output/homomer/N6_multimer_structure_generation/default_multimer
Cannot find default alphafold alignments for A:
/multicom_runs/output/homomer/N6_multimer_structure_generation/default_multimer/msas/A/m
onomer_final.a3m
Program failed in step 6
Multimer structure generation has been finished!
```

For heteromer, it crashes also at step 6.

Here is my input fasta file heteromer.fasta:

```
>test
PIAQIHILEGRSDEQKETLIREVSEAIRSLDAPLTSVRVIITEMAKGHFGIGGELASK
>test2
PIAQIHILLELELATEGRSDEQKETLIREVSKKHWWYWEAIRSLDAPLTSVRVIITEMAKGHFGIGGELASK
```

and the logs where it crashes:

```
I0927 22:23:00.082472 139675074123584 xla_bridge.py:353] Unable to initialize backend
'tpu_driver': NOT_FOUND: Unable to find driver in registry given worker:
I0927 22:23:00.252418 139675074123584 xla_bridge.py:353] Unable to initialize backend
'rocm': NOT_FOUND: Could not find registered platform with name: "rocm". Available platform
names are: Host Interpreter CUDA
I0927 22:23:00.253169 139675074123584 xla_bridge.py:353] Unable to initialize backend 'tpu':
module 'jaxlib.xla_extension' has no attribute 'get_tpu_client'
I0927 22:23:00.253403 139675074123584 xla_bridge.py:353] Unable to initialize backend
'plugin': xla_extension has no attributes named get_plugin_device_client. Compile TensorFlow with
//tensorflow/compiler/xla/python:enable_plugin_device set to true (defaults to false) to enable
this.
I0927 22:23:06.122310 139675074123584 run_alphafold_multimer_custom.py:413] Have 25
models: ['model_1_multimer_v3_pred_0', 'model_1_multimer_v3_pred_1',
'model_1_multimer_v3_pred_2', 'model_1_multimer_v3_pred_3', 'model_1_multimer_v3_pred_4',
'model_2_multimer_v3_pred_0', 'model_2_multimer_v3_pred_1', 'model_2_multimer_v3_pred_2',
'model_2_multimer_v3_pred_3', 'model_2_multimer_v3_pred_4', 'model_3_multimer_v3_pred_0',
'model_3_multimer_v3_pred_1', 'model_3_multimer_v3_pred_2', 'model_3_multimer_v3_pred_3',
'model_3_multimer_v3_pred_4', 'model_4_multimer_v3_pred_0', 'model_4_multimer_v3_pred_1',
'model_4_multimer_v3_pred_2', 'model_4_multimer_v3_pred_3', 'model_4_multimer_v3_pred_4',
'model_5_multimer_v3_pred_0', 'model_5_multimer_v3_pred_1', 'model_5_multimer_v3_pred_2',
'model_5_multimer_v3_pred_3', 'model_5_multimer_v3_pred_4']
I0927 22:23:06.122452 139675074123584 run_alphafold_multimer_custom.py:417] Using
random seed 259507381966910969 for the data pipeline
I0927 22:23:06.122716 139675074123584 run_alphafold_multimer_custom.py:154] Predicting
heteromer
I0927 22:23:06.123090 139675074123584 templates_custom.py:967] Searching for template
for:
PIAQIHILEGRSDEQKETLIREVSEAIRSLDAPLTSVRVIITEMAKGHFGIGGELASK,PIAQIHILLELELATEGRS
DEQKETLIREVSKKHWWYWEAIRSLDAPLTSVRVIITEMAKGHFGIGGELASK
False
Traceback (most recent call last):
File "run_alphafold_multimer_custom.py", line 467, in <module>
app.run(main)
File "/software/miniconda3/envs/multicom-2.0.2/lib/python3.8/site-packages/absl/app.py", line
312, in run
_run_main(main, args)
File "/software/miniconda3/envs/multicom-2.0.2/lib/python3.8/site-packages/absl/app.py", line
258, in _run_main
sys.exit(main(argv))
```

```
File "run_alphafold_multimer_custom.py", line 446, in main
predict_structure(
File "run_alphafold_multimer_custom.py", line 168, in predict_structure
feature_dict = data_pipeline.process(
File "/Utils/MULTICOM3/alphafold-2.3.2/alphafold/data_custom/pipeline_multimer_custom.py",
line 431, in process
monomer_models_temp_results =
self.monomer_template_featurizer.get_templates(chain_id_map,
File "/Utils/MULTICOM3/alphafold-2.3.2/alphafold/data_custom/templates_custom.py", line 979, in
get_templates
template_chain=chain_id_map[chainid].description[4],
IndexError: string index out of range
```

20) OK

21) OK

22) OK

23) If I understand well, it is not possible to avoid recomputing the MSA if this step was already done by Multicom, is it correct ?

My question was also to know how to specify other parameters for AlphaFold, especially "models_to_relax" and "template_mmcif_dir" ?

24) OK

25) I still got a few "/" in the logs for homomers and heteromers (for hhsearch, hmmbuild, foldseek)

Response to Review Comments

Reviewer #2

The authors answered to all my comments. I reply here to their answers, "OK" meaning I don't have anything else to reply.

Response:

Thank you very much for carefully checking our revisions of the last round and provided valuable critiques for us to further improve the manuscript. Below are our responses to your comments that were not sufficiently addressed last time.

1) The 4 first references are not really appropriate in the context of protein-protein docking and references connected to protein-protein docking should replace them.

Response:

Thank you for the valuable suggestion. We have rewritten the paragraph and replaced the first 4 references with new references 1 to 7 as follows:

In contrast, ab-initio methods aim to predict complex structures without the reliance on templates, utilizing various techniques such as grid-based fast-Fourier transform docking¹⁻³, particle swarm optimization for conformational sampling⁴, and local shape complementarity alongside symmetry constraints⁵. Furthermore, integrative methods^{6,7} combine template-based modeling and ab-initio docking to improve the prediction of complex structures. Nevertheless, the accuracy of these methods for predicting complex structures is generally low^{8,9}, due to the absence of templates, complexities associated with conformational sampling, inaccuracy of scoring functions, and challenges in accommodating protein flexibility.

New references:

1. Kozakov D, Hall DR, Xia B, et al. The ClusPro web server for protein–protein docking. *Nature protocols* 2017;12:255-278.
2. Pierce BG, Wiehe K, Hwang H, et al. ZDOCK server: interactive docking prediction of protein–protein complexes and symmetric multimers. *Bioinformatics* 2014;30:1771-1773.
3. Macindoe G, Mavridis L, Venkatraman V, et al. HexServer: an FFT-based protein docking server powered by graphics processors. *Nucleic acids research* 2010;38:W445-W449.
4. Torchala M, Moal IH, Chaleil RA, et al. SwarmDock: a server for flexible protein–protein docking. *Bioinformatics* 2013;29:807-809.
5. Schneidman-Duhovny D, Inbar Y, Nussinov R, et al. PatchDock and SymmDock: servers for rigid and symmetric docking. *Nucleic acids research* 2005;33:W363-W367.
6. Yan Y, Tao H, He J, et al. The HDock server for integrated protein–protein docking. *Nature protocols* 2020;15:1829-1852.
7. Duan R, Qiu L, Xu X, et al. Performance of human and server prediction in CAPRI rounds 38-45. *Proteins: Structure, Function, and Bioinformatics* 2020;88:1110-1120.

8. Lensink MF, Brysbaert G, Nadzirin N, et al. Blind prediction of homo-and hetero-protein complexes: The CASP13-CAPRI experiment. *Proteins: Structure, Function, and Bioinformatics* 2019;87:1200-1221.

9. Lensink MF, Brysbaert G, Mauri T, et al. Prediction of protein assemblies, the next frontier: The CASP14-CAPRI experiment. *Proteins: Structure, Function, and Bioinformatics* 2021;89:1800-1823.

4) I would slightly moderate “For DockQ score that specifically considers the quality of the interface”, adding a “more” before “specifically”, considering the LRMS part of the DockQ computation.

Response:

Thank you for the great suggestion. We have made the changes accordingly.

13) Thanks for details of the configurations. My question was however rather to know if the authors recommend a minimum amount in RAM in particular.

Response:

Thank you for raising the question. We recommend a minimum of 85GB of RAM to run the MULTICOM3 system. We added this information into the readme file at the GitHub repository.

Github repository :

19) But for homomers and heteromers, it crashes at some point. For homomers, it crashes at step 6:

Here is my input fasta file homomer.fasta:

```
>test
```

```
PIAQIHILEGRSDEQKETLIREVSEAIRSLDAPLTSVRVIITEMAKGHFGIGGELASK
```

```
>test
```

```
PIAQIHILEGRSDEQKETLIREVSEAIRSLDAPLTSVRVIITEMAKGHFGIGGELASK
```

and the logs where it crashes:

```
python run_alphafold_pre.py --fasta_path /multicom_runs/input/homomer.fasta --bfd_uniref_a3ms  
/multicom_runs/output/homomer/N1_monomer_alignments_generation/A/A_uniref30_bfd.a3m,/m  
ulticom_runs/output/homomer/N1_monomer_alignments_generation/B/B_uniref30_bfd.a3m --  
mgnify_stos  
/multicom_runs/output/homomer/N1_monomer_alignments_generation/A/A_mgnify.sto,/multicom  
_runs/output/homomer/N1_monomer_alignments_generation/B/B_mgnify.sto --uniref90_stos  
/multicom_runs/output/homomer/N1_monomer_alignments_generation/A/A_uniref90.sto,/multico  
m_runs/output/homomer/N1_monomer_alignments_generation/B/B_uniref90.sto --uniprot_stos  
/multicom_runs/output/homomer/N1_monomer_alignments_generation/A/A_uniprot.sto,/multico  
m_runs/output/homomer/N1_monomer_alignments_generation/B/B_uniprot.sto --env_dir  
/miniconda3/envs/multicom-2.0.2/bin/ --database_dir /shared/databases/2023-07-31 --
```

```
num_multimer_predictions_per_model 5 --multimer_num_ensemble 1 --multimer_num_recycle 3 --
output_dir /multicom_runs/output/homomer/N6_multimer_structure_generation/default_multimer
Cannot find default alphafold alignments for A:
/multicom_runs/output/homomer/N6_multimer_structure_generation/default_multimer/msas/A/m
onomer_final.a3m
Program failed in step 6
Multimer structure generation has been finished!
```

For heteromer, it crashes also at step 6.

Here is my input fasta file heteromer.fasta:

```
>test
PIAQIHILEGRSDEQKETLIREVSEAIRSLDAPLTSVRVIITEMAKGHFGIGGELASK
>test2
PIAQIHILLELELATEGRSDEQKETLIREVSKKHWWYWEAIRSLDAPLTSVRVIITEMAKGHFGIGGELASK
```

and the logs where it crashes:

```
I0927 22:23:00.082472 139675074123584 xla_bridge.py:353] Unable to initialize backend 'tpu_driver':
NOT_FOUND: Unable to find driver in registry given worker:
I0927 22:23:00.252418 139675074123584 xla_bridge.py:353] Unable to initialize backend 'rocm':
NOT_FOUND: Could not find registered platform with name: "rocm". Available platform names are:
Host Interpreter CUDA
I0927 22:23:00.253169 139675074123584 xla_bridge.py:353] Unable to initialize backend 'tpu': module
'jaxlib.xla_extension' has no attribute 'get_tpu_client'
I0927 22:23:00.253403 139675074123584 xla_bridge.py:353] Unable to initialize backend 'plugin':
xla_extension has no attributes named get_plugin_device_client. Compile TensorFlow with
//tensorflow/compiler/xla/python:enable_plugin_device set to true (defaults to false) to enable this.
I0927 22:23:06.122310 139675074123584 run_alphafold_multimer_custom.py:413] Have 25 models:
['model_1_multimer_v3_pred_0', 'model_1_multimer_v3_pred_1', 'model_1_multimer_v3_pred_2',
'model_1_multimer_v3_pred_3', 'model_1_multimer_v3_pred_4', 'model_2_multimer_v3_pred_0',
'model_2_multimer_v3_pred_1', 'model_2_multimer_v3_pred_2', 'model_2_multimer_v3_pred_3',
'model_2_multimer_v3_pred_4', 'model_3_multimer_v3_pred_0', 'model_3_multimer_v3_pred_1',
'model_3_multimer_v3_pred_2', 'model_3_multimer_v3_pred_3', 'model_3_multimer_v3_pred_4',
'model_4_multimer_v3_pred_0', 'model_4_multimer_v3_pred_1', 'model_4_multimer_v3_pred_2',
'model_4_multimer_v3_pred_3', 'model_4_multimer_v3_pred_4', 'model_5_multimer_v3_pred_0',
'model_5_multimer_v3_pred_1', 'model_5_multimer_v3_pred_2', 'model_5_multimer_v3_pred_3',
'model_5_multimer_v3_pred_4']
```

I0927 22:23:06.122452 139675074123584 run_alphafold_multimer_custom.py:417] Using random seed 259507381966910969 for the data pipeline

I0927 22:23:06.122716 139675074123584 run_alphafold_multimer_custom.py:154] Predicting heteromer

I0927 22:23:06.123090 139675074123584 templates_custom.py:967] Searching for template for: PIAQIHILEGRSDEQKETLIREVSEAIRSLDAPLTSVRVIITEMAKGHFGIGGELASK,PIAQIHILLELELATEGRSDEQKETLIREVSKKHWWYWEAIRSLDAPLTSVRVIITEMAKGHFGIGGELASK

False

Traceback (most recent call last):

File "run_alphafold_multimer_custom.py", line 467, in <module>

app.run(main)

File "/software/miniconda3/envs/multicom-2.0.2/lib/python3.8/site-packages/absl/app.py", line 312, in run

_run_main(main, args)

File "/software/miniconda3/envs/multicom-2.0.2/lib/python3.8/site-packages/absl/app.py", line 258, in _run_main

sys.exit(main(argv))

File "run_alphafold_multimer_custom.py", line 446, in main

predict_structure(

File "run_alphafold_multimer_custom.py", line 168, in predict_structure

feature_dict = data_pipeline.process(

File "/Utils/MULTICOM3/alphafold-2.3.2/alphafold/data_custom/pipeline_multimer_custom.py", line 431, in process

monomer_models_temp_results = self.monomer_template_featurizer.get_templates(chain_id_map,

File "/Utils/MULTICOM3/alphafold-2.3.2/alphafold/data_custom/templates_custom.py", line 979, in get_templates

template_chain=chain_id_map[chainid].description[4],

IndexError: string index out of range

Response:

Thank you for bringing the issue to our attention. Both problems are caused by the sequence name in the fasta file. Typically, we assign distinct sequence names along with their chain IDs in the fasta file, for instance, T1109_A and T1109_B, or H1106_A and H1106_B. We did not test the other cases for the input fasta file. In the latest version of MULTICOM3 (v2.1.0), we have fixed the bugs and successfully run the MULTICOM3 system with your heteromer and homomer fasta file as input. Please refer to the instructions in <https://github.com/BioinfoMachineLearning/MULTICOM3> to update the MULTICOM3 system, as we have provided new installation settings and database files. For example, we have introduced additional customizable parameters for running AlphaFold2/AlphaFold-Multimer, and a more user-friendly Docker version of MULTICOM3. You can also run the following steps to update the MULTICOM3 using the non-docker installation (where YOUR_MULTICOM3_DB_DIR in the commands should be replaced with your \$MULTICOM3_INSTALL_DIR/databases):

1. Run the command: `python download_database_and_tools.py --multicom3db_dir <YOUR_MULTICOM3_DB_DIR>`
2. Run the command: `python configure.py --envdir ~/miniconda3/envs/multicom3 --multicom3db_dir <YOUR_MULTICOM3_DB_DIR> --afdb_dir <YOUR_ALPHAFOLD_DB_DIR>`

23) If I understand well, it is not possible to avoid recomputing the MSA if this step was already done by Multicom, is it correct ? My question was also to know how to specify other parameters for AlphaFold, especially “models_to_relax” and “template_mmcif_dir” ?

Response:

Thank you for good questions. We can set the parameter “use_precomputed_msas” to “True” if we have already generated the MSAs prior to generating the predictions. By setting the parameter, we can let AlphaFold2/AlphaFold-Multimer to read the generated MSAs to avoid recomputing them.

To specify the parameters for running AlphaFold2/AlphaFold-Multimer, you can adjust the values for some of the parameters in the db_option file. Here is the table of the input parameters for the default version of AlphaFold2/AlphaFold-Multimer (name of the parameter, whether the parameter is fixed, name of the parameter in the db_option file, default value and note):

Parameters	Fixed	Name in the db_option file	Default value	Note
jackhmmmer_binary_path	Yes	/	\$YOUR_ENV/jackhmmmer	These parameters are fixed because the parameters are automatically generated during the installation.
hhblits_binary_path	Yes	/	\$YOUR_ENV/hhblits	
hhsearch_binary_path	Yes	/	INSTALLDIR_TOOLS/hhsuite-3.2.0/bin/hhsearch	
hmmsearch_binary_path	Yes	/	\$YOUR_ENV/hmmsearch	
hmmbuild_binary_path	Yes	/	\$YOUR_ENV/hmmbuild	
kalign_binary_path	Yes	/	\$YOUR_ENV/kalign	
data_dir	Yes	/	\$YOUR_ALPHAFOLD_DB_DIR	These parameters are fixed because the parameters are automatically generated during the installation. However, you can replace the files in the path if you want to use different sources of the databases.
uniref90_database_path	Yes	/	\$YOUR_ALPHAFOLD_DB_DIR/uniref90	
mgnify_database_path	Yes	/	\$YOUR_ALPHAFOLD_DB_DIR/mgnify	
bfd_database_path	Yes	/	\$YOUR_ALPHAFOLD_DB_DIR/bfd	
uniref30_database_path	Yes	/	\$YOUR_ALPHAFOLD_DB_DIR/uniref30	
uniprot_database_path	Yes	/	\$YOUR_ALPHAFOLD_DB_DIR/uniprot	
pdb70_database_path	Yes	/	\$YOUR_ALPHAFOLD_DB_DIR/pdb70	
pdb_seqres_database_path	Yes	/	\$YOUR_ALPHAFOLD_DB_DIR/pdb_seqres	

template_mmcif_dir	Yes	/	\$YOUR_ALPHAFOLD_DB_DIR/pdb_mmcif/mmcif_files	
obsolete_pdbs_path	Yes	/	\$YOUR_ALPHAFOLD_DB_DIR/pdb_mmcif/obsolete.dat	
max_template_date	No	max_template_date	2024-06-01	You can change the parameters in the db_option file
model_preset	No	monomer_model_preset / multimer_model_preset	monomer / multimer	
Benchmark	No	alphafold_benchmark	True	
num_multimer_predictions_per_model	No	num_multimer_predictions_per_model	5	
models_to_relax	No	models_to_relax	ALL	
use_gpu_relax	No	use_gpu_relax	True	
db_preset	Yes	/	full_dbs	Always be 'full_dbs'
small_bfd_database_path	Yes	/	/	Always be empty because the db_preset is always 'full_dbs'
random_seed	Yes	/	/	Always randomly generated
use_precomputed_msas	Yes	/	True	Always be 'True'

Although some of the parameters are fixed, you can replace the file/folder of the specified paths in the db_option file to specify other values for the fixed parameters.

Compared to MULTICOM3 v2.0.2, we have provided more customized parameters in the latest version of MULTICOM3 (v2.1.0) for running AlphaFold2/AlphaFold-Multimer, for example max_template_date, model_preset, benchmark, models_to_relax, and use_gpu_relax.

25) I still got a few “//” in the logs for homomers and heteromers (for hhsearch, hmmbuild, foldseek)

Response:

Thank you for finding the problem. We have resolved the problem in the latest version of MULTICOM3 (v2.1.0) on GitHub.